# Functional diversity of dopamine axons in prefrontal cortex during classical conditioning

**Kenta Abe[1†], Yuki Kambe[2†], Kei Majima[3,4†], Zijing Hu[5,6†], Makoto Ohtake[1†], Ali Momennezhad[2], Hideki Izumi[7], Takuma Tanaka[7], Ashley Matunis[1,8,9], Emma Stacy[1,8], Takahide Itokazu[9], Takashi R Sato[1*‡], Tatsuo Sato[2,4,5,6,10]\***

[1]Department of Neuroscience, Medical University of South Carolina, Charleston, United States; [2]Department of Pharmacology, Kagoshima University, Kagoshima, Japan; [3]Institute for Quantum Life Science, National Institutes for Quantum Science and Technology, Chiba, Japan; [4]Japan Science and Technology PRESTO, Saitama, Japan; [5]Department of Physiology, Monash University, Clayton, Australia; [6]Neuroscience Program, Biomedicine Discovery Institute, Monash University, Clayton, Australia; [7]Faculty of Data Science, Shiga University, Shiga, Japan; [8]Department of Biology, College of Charleston, Charleston, United States; [9]Department of Neuro-Medical Science, Osaka University, Osaka, Japan; [10]Japan Science and Technology FOREST, Saitama, Japan

**\*For correspondence:**
satot@musc.edu (TRS);
tatsuo.sato@m.kufm.kagoshima-u.ac.jp (TS)

†These authors contributed equally to this work

‡Lead contact.

**Competing interest:** The authors declare that no competing interests exist.

**Abstract** Midbrain dopamine neurons impact neural processing in the prefrontal cortex (PFC) through mesocortical projections. However, the signals conveyed by dopamine projections to the PFC remain unclear, particularly at the single-axon level. Here, we investigated dopaminergic axonal activity in the medial PFC (mPFC) during reward and aversive processing. By optimizing microprism-mediated two-photon calcium imaging of dopamine axon terminals, we found diverse activity in dopamine axons responsive to both reward and aversive stimuli. Some axons exhibited a preference for reward, while others favored aversive stimuli, and there was a strong bias for the latter at the population level. Long-term longitudinal imaging revealed that the preference was maintained in reward- and aversive-preferring axons throughout classical conditioning in which rewarding and aversive stimuli were paired with preceding auditory cues. However, as mice learned to discriminate reward or aversive cues, a cue activity preference gradually developed only in aversive-preferring axons. We inferred the trial-by-trial cue discrimination based on machine learning using anticipatory licking or facial expressions, and found that successful discrimination was accompanied by sharper selectivity for the aversive cue in aversive-preferring axons. Our findings indicate that a group of mesocortical dopamine axons encodes aversive-related signals, which are modulated by both classical conditioning across days and trial-by-trial discrimination within a day.

## eLife assessment

This **important** study shows that distinct midbrain dopaminergic axons in the medial prefrontal cortex respond to aversive and rewarding stimuli and suggest that they are biased toward aversive processing. The use of innovative microprism based two-photon calcium imaging to study single axon heterogeneity is **convincing**, although the experimental design makes it difficult to definitively distinguish aversive valence from stimulus salience in this dopamine projection. This work will be of interest to neuroscientists working on neuromodulatory systems, cortical function and decision making.

## Introduction

The prefrontal cortex (PFC) contributes to a variety of higher cognitive functions, achieving the flexible control of behaviors that enables animals to adapt to a changing environment (*Miller and Cohen, 2001*; *Fuster, 2015*). The PFC is involved, for instance, in stimulus selection, working memory, rule switching, and decision making (*Miller and Wallis, 2009*). PFC processing and circuits are highly sensitive to neuromodulators, including dopamine (*Seamans and Yang, 2004*; *Arnsten et al., 2012*). Indeed, studies using pharmacological or optogenetic manipulation of dopamine signaling have suggested roles of dopamine in gating sensory signals (*Popescu et al., 2016*; *Vander Weele et al., 2018*), maintaining working memory (*Sawaguchi and Goldman-Rakic, 1994*), and relaying decisions to motor structures (*Ott et al., 2014*). Consistently, dysregulation of dopamine signaling in the PFC has been suggested to underlie a wide array of neuropsychiatric disorders, including schizophrenia, depression, attention-deficit/hyperactivity disorder, and post-traumatic stress disorder (*Okubo et al., 1997*; *Lindström et al., 1999*; *Granon et al., 2000*; *Arnsten and Dudley, 2005*; *Howes and Kapur, 2009*; *Hoexter et al., 2012*; *Grace, 2016*).

The PFC receives dopaminergic inputs from a subset of dopamine neurons in the midbrain, but the information encoded by this subset in vivo remains unclear. Decades of investigations have revealed that midbrain dopamine neurons in the ventral tegmental area (VTA) generally encode reward prediction errors (*Schultz et al., 1997*): the neurons increase their firing to unexpected reward delivery and shift their response to cues that precede reward delivery after instrumental learning or classical conditioning (*Rescorla and Wagner, 1972*; *Sutton and Barto, 1981*). However, several studies have reported that a subpopulation of dopamine neurons show phasic responses to aversive stimuli as a part of salience signaling (*Chiodo et al., 1980*; *Mantz et al., 1989*; *Guarraci and Kapp, 1999*; *Matsumoto and Hikosaka, 2009*), implying that midbrain dopamine neurons may not be functionally homogeneous. Indeed, depending on the projection target, dopamine neurons can have distinct molecular, anatomical, and electrophysiological features (*Lammel et al., 2008*; *Poulin et al., 2018*). Dopamine neurons that project to the PFC might locate primarily to the medial posterior VTA (*Lammel et al., 2008*) and show different genetic profiles from other dopamine neurons (*Poulin et al., 2018*). In addition, optogenetic stimulation of the PFC-projecting DA neurons does not reinforce specific actions (*Popescu et al., 2016*; *Ellwood et al., 2017*; *Vander Weele et al., 2018*). Moreover, these neurons might respond not only to rewarding stimuli but also to aversive stimuli. Microdialysis, amperometry, and voltammetry measurements in the PFC have demonstrated an increase of dopamine in response to appetitive stimuli (*Hernandez and Hoebel, 1990*; *Ahn and Phillips, 1999*; *St. Onge et al., 2012*), aversive stimuli (*Thierry et al., 1976*; *Abercrombie et al., 1989*; *Finlay et al., 1995*; *Vander Weele et al., 2018*), or both (*Bassareo et al., 2002*). Similarly, measurements of the bulk calcium activity of mesocortical dopaminergic fibers have shown responses to appetitive (*Ellwood et al., 2017*) and aversive (*Kim et al., 2016*) stimuli. This apparent discrepancy is difficult to reconcile because none of these approaches could investigate the activity of individual dopamine neurons. Moreover, most previous studies evaluated the effects of either a rewarding or an aversive stimulus, rather than both. Consequently, it remains unknown whether the same or different mesocortical dopamine neurons respond to behaviorally opposing stimuli. It is also not known how these dopamine neurons change their response during classical conditioning, where rewarding or aversive stimuli are paired with conditioned cues.

To address these knowledge gaps, we developed an approach for imaging individual dopamine axons based on in vivo two-photon imaging with a microprism (*Low et al., 2014*). We optimized the microprism design and imaged dopamine axon terminals expressing genetically encoded calcium sensors in the mouse medial PFC (mPFC). We then head-fixed the mice to give rewards or aversive stimuli (water drops or electrical shocks) and trained the mice to associate the stimuli with preceding auditory cues (classical conditioning). During classical conditioning, we tracked the activity of dopamine axons over a period of days. We found that the dopamine axons showed diverse preferences for unconditioned (rewarding or aversive) stimuli. Through the classical conditioning, activity preferences for conditioned auditory cues were enhanced only for aversive-preferring axons. Moreover, in aversive-preferring axons, a machine learning-based analysis revealed that cue activity became more selective when the behavior of animals was judged as correct. We conclude that mesocortical dopamine axon activity is involved in aversive-related processing that is modulated by both classical conditioning across days and trial-by-trial judgements of conditioned cues within a day.

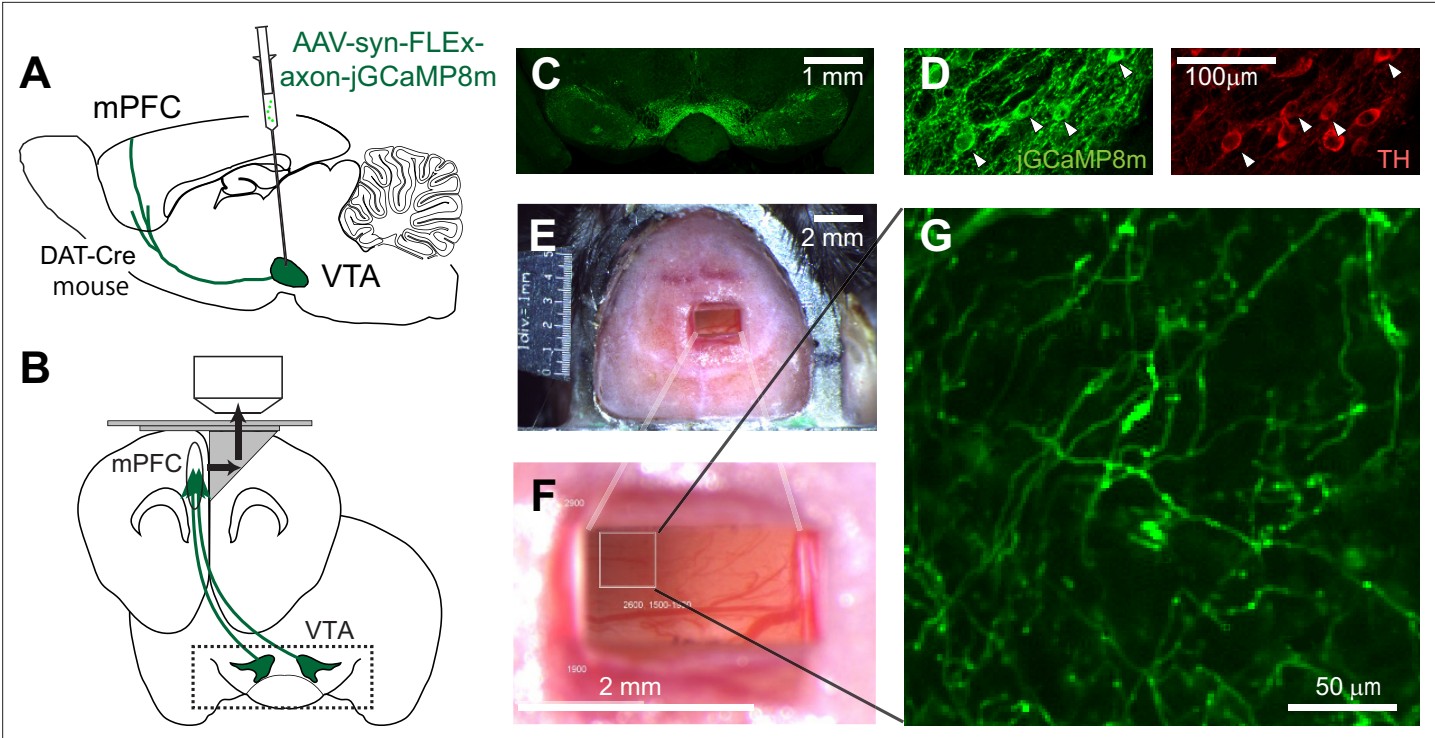

**Figure 1.** Two-photon imaging of dopaminergic axons projecting to the mPFC. (**A**, **B**) Experimental design. The activity of midbrain dopamine neurons projecting to the mPFC was measured by two-photon calcium imaging of their axons. The axons were accessed through a microprism that bends the optical axis inside the brain (black arrows in B). (**C**) GCaMP was expressed virally in dopamine neurons in DAT-Cre transgenic mice. A coronal section shows GCaMP expression in the VTA, demonstrating that AAV-axon-DIO-jGCaMP8m was injected into the VTA. (**D**) jGCaMP8m-expressing neurons were positive for tyrosine hydroxylase (TH), a marker for dopamine neurons. (**E**, **F**) Dorsal view of a mouse head implanted with a microprism assembly. The microprism was 1x2 mm. (**G**) An example in vivo image of jGCaMP8m-expressing axons.

The online version of this article includes the following figure supplement(s) for figure 1:

**Figure supplement 1.** Sparse dopaminergic projections to the the mPFC.

**Figure supplement 2.** Double-layer glass significantly reduces brain movement.

**Figure supplement 3.** Long-term imaging of dopamine axons across days.

## Results

### Two-photon imaging shows dopaminergic axons in the mPFC of awake mice

To investigate the signal sent by dopamine neurons to the mPFC in mice, we developed an approach based on two-photon imaging using a microprism (*Andermann et al., 2013*; *Low et al., 2014*). We first expressed axon-jGCaMP8m, an axon-targeted (*Broussard et al., 2018*) genetically encoded calcium sensor (*Zhang et al., 2023*), in dopamine neurons in the VTA. We injected Cre-dependent AAV into the midbrain regions of transgenic mice (DAT-Cre), which express Cre-recombinase in dopamine neurons (*Kim et al., 2016*; *Figure 1A*, see Materials and methods). After 2–3 weeks, using sectioned slices, we confirmed that GCaMP expression in cell bodies in the VTA (and substantia nigra pars compacta [SNc]) (*Figure 1C*) coincides with the expression of tyrosine hydroxylase, an endogenous marker for dopamine neurons (*Figure 1D*). Dopamine neurons in the VTA are known to project sparsely to the mPFC, including the superficial layers (*Vander Weele et al., 2018*; *Figure 1—figure supplement 1*), but the mPFC itself is located deep in the medial bank (*Figure 1B*), rendering two-photon imaging of GCaMP (which is typically excited at 920–980 nm) infeasible. Therefore, we inserted a microprism into the longitudinal fissure between the two medial banks (two hemispheres) to optically access the mPFC (*Figure 1E–F*). The right-angle microprism bends the optical axis within the brain, providing optical access to the fissure wall and the mPFC surface (*Low et al., 2014*). We optimized the microprism assembly (*Figure 1—figure supplement 2C*) in order to reach up to 2 mm

in depth from the dorsal surface (*Figure 1F*). The assembly incorporated double-layer glass at the top (*Komiyama et al., 2010*), stabilizing the brain from both the medial and dorsal sides, which significantly reduced the movement of the brain (*Figure 1—figure supplement 2*). Through the microprism, we could visualize GCaMP-expressing axons in the superficial layers of the mPFC in live animals (at a depth of 30–100 μm, *Figure 1G*, *Figure 1—figure supplement 3*). Sparse axons in the superficial layers are advantageous for two-photon imaging, achieving low background noise. In contrast, axons in the deep layers, which are known to be denser in sectioned slices (*Figure 1—figure supplement 3*), could not be visualized under our experiments. The GCaMP signal can indicate the calcium influx into axons and terminals, which is triggered by axonal action potentials (*Petreanu et al., 2012*; *Howe and Dombeck, 2016*; *Lutas et al., 2019*), thereby providing a measure of the activity of dopamine neurons that send projections to the mPFC. In contrast, when we inserted a gradient refractive index (GRIN) lens into the mPFC (*Kamigaki and Dan, 2017*), we could not reliably visualize GCaMP-expressing dopamine axons, unlike the case for dopamine axons in the basal amygdala (*Lutas et al., 2019*). This difference might indicate that the dopamine axons in the mPFC have weaker signals requiring a lens with a larger numerical aperture (GRIN lens: NA 0.5 vs. Nikon objective lens: NA 0.8) or that these axons are less resilient to mechanical damage in close vicinity.

## Dopaminergic axons in the mPFC have diverse responses to rewarding and aversive stimuli

Using our imaging approach, we first investigated whether individual dopamine axons respond to unexpected rewards (*Schultz et al., 1997*) and unexpected aversive stimuli (*Vander Weele et al., 2018*). As a reward, we delivered drops of water through a spout with random timing to water-deprived mice (*Figure 2A*). In response to the reward delivery, the mice licked the water spout, and we filmed this behavior to quantify the tongue position (*Figure 2B* and *Figure 2—figure supplement 1*). Upon the delivery of a water drop, the mice started licking (licking latency: 0.538±0.065 s, n=8 animals; *Otis et al., 2017*). Two-photon calcium imaging revealed that the water reward evoked brief calcium transients in many dopamine axons (40.1% of axons in eight animals, example in *Figure 2I*). The brief calcium response to the reward is consistent with increased phasic firing in dopamine neurons at the time of unexpected reward, as previously reported in many studies in primates (*Schultz et al., 1997*) and rodents (*Engelhard et al., 2019*; *Amo et al., 2022*).

In contrast to conventional midbrain dopamine neurons, mPFC dopamine axons are proposed to play a key role in aversive processing (*Weele et al., 2019*). To investigate the calcium response to an unexpected aversive stimulus, we delivered mild electrical shocks to the tail of the mice (*Kim et al., 2016*; *Patriarchi et al., 2018*; *Lutas et al., 2019*; *Figure 2A*) that were randomly interleaved with reward delivery (one shock for every seven rewards on average). The mild shock evoked calcium transients in many dopamine projections (*Figure 2F, G and H*), together with locomotion (*Figure 2B*). These transients could simply reflect locomotion initiation, similar to dopamine axons in the dorsal striatum (*Howe and Dombeck, 2016*). To explore this possibility, we investigated whether locomotion without aversive stimuli is accompanied by increased calcium activity. We found no significant calcium increase at the time point of spontaneous locomotion initiation (*Figure 2—figure supplement 2*). Therefore, unlike the axons projecting from the SNc to the dorsal striatum (*Howe and Dombeck, 2016*; *Ma et al., 2022*), mPFC dopamine axons do not encode the initiation of movement; rather, these axons respond to the aversive stimulus.

Some previous studies have demonstrated that the overall dopamine release at the mPFC or the summed activity of mPFC dopamine axons exhibits a strong response to aversive stimuli (e.g. tail shock), but little to rewards (*Kim et al., 2016*). We evaluated the preference of individual axons for rewarding and aversive signals at a single-axon resolution by computing the polar angle for individual axons on a Cartesian representation of reward and shock activity (*Figure 2J and K*, *Figure 2—figure supplement 3*). In the polar representation, an angle of 0° indicates a strong preference for reward information, whereas 90° indicates a preference for aversive information. The polar angle distribution revealed that a significant number of axons preferred aversive stimuli, although some preferred reward. As a result, probability density, estimated by kernel smoothing, showed a bimodal distribution (*Figure 2K*, solid line) with a trough at around 45–50°. In addition, axons showing significant responses were categorized into two clusters based on k-means clustering (*Figure 2J*), the separation of which coincided roughly with 45–50° (*Figure 2K*). Thereafter, we refer to these clusters as

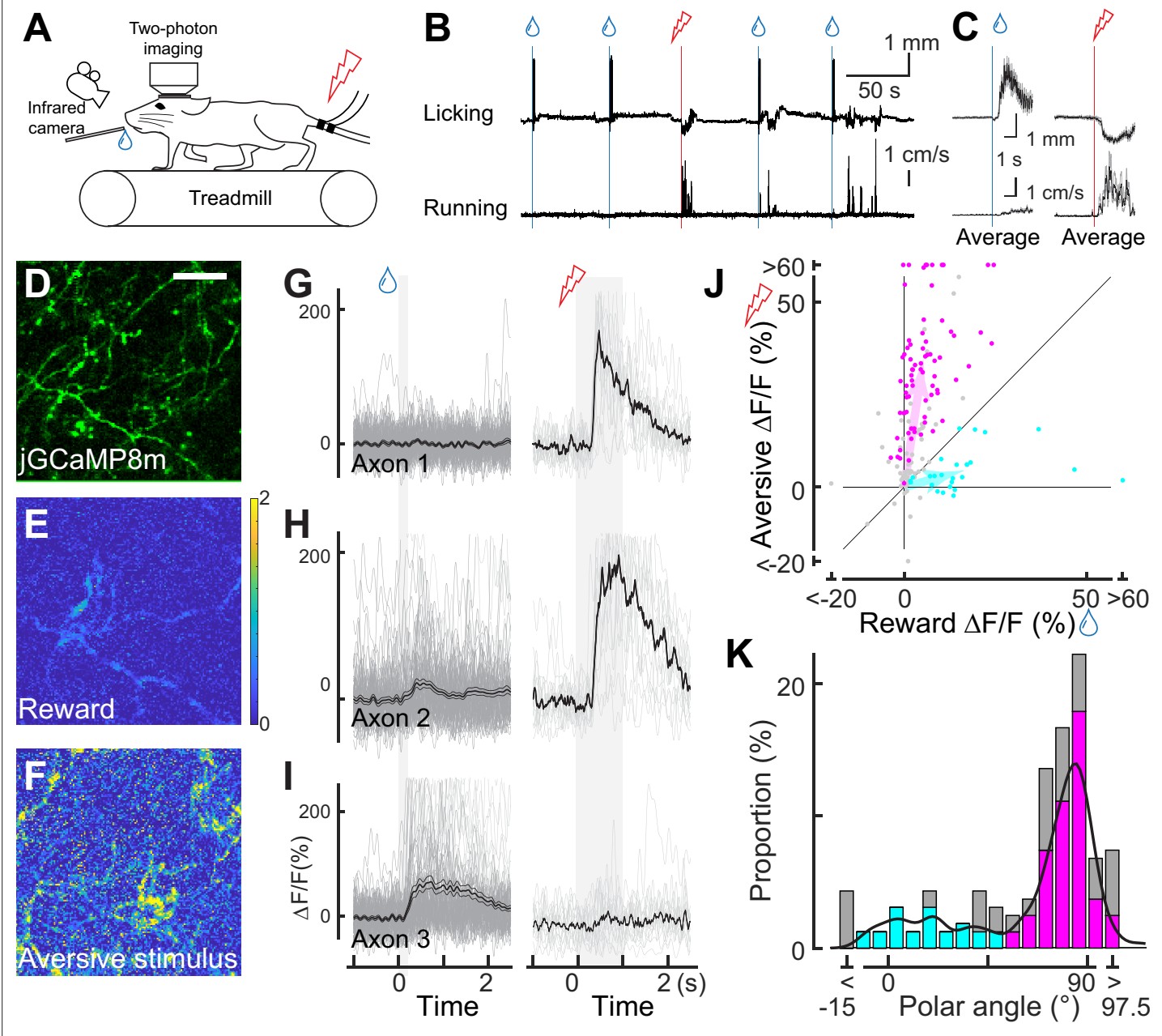

**Figure 2.** Dopaminergic axonal response to unexpected rewarding or aversive stimuli. (**A**) Experimental design. Mice were placed under a two-photon microscope on a linear treadmill and were given unexpected rewarding (water drops) or aversive (electrical shock to the tail) stimuli. The mouse's face was filmed with an infrared camera to track the tongue position. (**B**) Example behavioral response to rewarding (top, tongue position) or aversive (bottom, treadmill speed) stimuli. (**C**) Average behavioral response on a single day for the same animal shown in B. (**D**) Representative image for dopamine axons labeled with jGCaMP8m. Scale bar: 50 μm. (**E, F**) Heatmaps of rewarding (**E**) or aversive (**F**) stimuli for the same imaging plane shown in D. (**G–I**) Calcium response for rewarding (left) and aversive (right) stimuli of three example axons. (**J**) Comparison between reward (x-axis) and aversive (y-axis) responses for dopamine axons (n=162). Statistically significant axons were labeled in either cyan (reward-preferring axons, n=25) or magenta (aversive-preferring axons, n=75). Vector averages representing reward-preferring axons and aversive-preferring axons are depicted as cyan and magenta arrows. (**K**) Histogram of the polar angle of the scatter plot in J (n=162). The solid line indicates probability density, estimated by kernel smoothing. A value of 0° represents axons that solely prefer rewards, whereas 90° represents those that solely prefer aversive stimuli.

The online version of this article includes the following figure supplement(s) for figure 2:

**Figure supplement 1.** Lick detection using DeepLabCut.

**Figure supplement 2.** Insignificant locomotion activity of dopaminergic axons.

*Figure 2 continued on next page*

*Figure 2 continued*

**Figure supplement 3.** Comparison between reward (x-axis) and aversive (y-axis) responses for each mouse (n = 8 animals), similar to Figure 2J (for all animals).

**Figure supplement 4.** Dopaminergic axonal activity depends on the reward volume and shock current.

aversive- or reward-preferring axons (colored magenta or cyan, respectively). These clusters do not respond exclusively to one hedonic valence (rewarding or aversive stimuli), as evident from the broad angle distributions. We could not find any anatomical patterns for aversive- or reward-preferring axons. These axons were present in either half of the prism view (i.e. anterior or posterior; ventral or dorsal), implying no obvious functional projection patterns within the mPFC. We note that the strength of preference could be quantitatively changed. Indeed, we found that the reward response to 10 µL nearly reached saturation, but the aversive response could be further increased at a stronger current (*Figure 2—figure supplement 4*). Therefore, the exclusive preference for aversive stimuli observed in some studies might possibly be explained by a smaller reward volume and/or stronger aversive stimulus. It may also be possible that the relative frequency of the aversive stimulus could influence the aversive preference. Moreover, measured signals in these studies may arise from deep layers and be different from the superficial axons that we image. Altogether, our two-photon imaging revealed, for the first time, that individual axons in the superficial layers show diverse preferences for rewarding and aversive stimuli.

## Aversive cue processing is enhanced in aversive-preferring axons during classical conditioning

How do the reward and aversive activities of individual axons change while animals are learning that the reward and aversive events are preceded and predicted by sensory cues? This paradigm, known as classical conditioning, is a key framework for capturing learning-related changes in midbrain dopamine neurons (*Sutton and Barto, 1981*; *Schultz et al., 1997*) and mPFC neurons (*Takehara-Nishiuchi and McNaughton, 2008*; *Otis et al., 2017*). We presented mice with a 2 s pure tone as a conditioned stimulus (CS_reward and CS_aversive: 9 and 13 kHz, or 13 and 9 kHz), and then, after a 1 s delay, we presented either a rewarding or an aversive unconditioned stimulus (*Figure 3A*). Previous work has shown that mice are able to discriminate between two tones that differ by more than 7% (*de Hoz and Nelken, 2014*). Indeed the mice learned the contingency between the conditioned stimulus (tone) and the outcome (reward or electrical shock), which was reflected in changes in their behavior throughout this conditioning process (*Figure 3B and C*). To quantify such behavioral changes during learning, we separated the learning into three phases in addition to the first day (*Figure 3D–G*, for six phases, *Figure 3—figure supplement 1*). On the first day, the animals licked the water spout only after the reward was delivered (first day; *Figure 3B*). However, during the middle and late phases, animals gradually showed licking behavior even before the reward delivery, representing an anticipation of reward (*Figure 3D*). We observed this anticipatory licking more frequently after CS_reward than CS_aversive (*Figure 3D* vs. 3E, p=0.031 for the middle phase, p=0.031 for the late phase, Wilcoxon signed-rank test, n=6 animals), indicating that the animals behaviorally learned to discriminate the two conditioned stimulus tones. Similarly, running before the delivery of the unconditioned stimulus was more frequent after CS_aversive than CS_reward at the late phase (*Figure 3F* vs. 3 G, p=0.031, Wilcoxon signed-rank test, n=6 animals), again indicating that the two conditioned auditory cues were behaviorally discriminated. Therefore, as in previous studies, anticipatory licking (*Otis et al., 2017*) and anticipatory running (*Lutas et al., 2019*) can capture whether animals behaviorally discriminate conditioned cues in classical conditioning.

Through the classical conditioning paradigm, our long-term two-photon imaging revealed that aversive-preferring dopamine axons maintained their preference for the unconditioned response but enhanced their selectivity for the aversive cue activity (*Figure 3H*). We evaluated the activity change at the time of the unconditioned stimuli (US) throughout the learning process for aversive- and reward-preferring axons (*Figure 3J*, magenta and cyan, respectively). On the first day, aversive-preferring dopamine axons showed stronger activity for the aversive stimuli (*Figure 3J*, top, *Figure 3—figure supplement 2A*), similar to the response without classical conditioning (*Figure 2J*). Across learning, the activity for the rewarding and aversive unconditioned stimuli gradually decreased (*Figure 3K*,

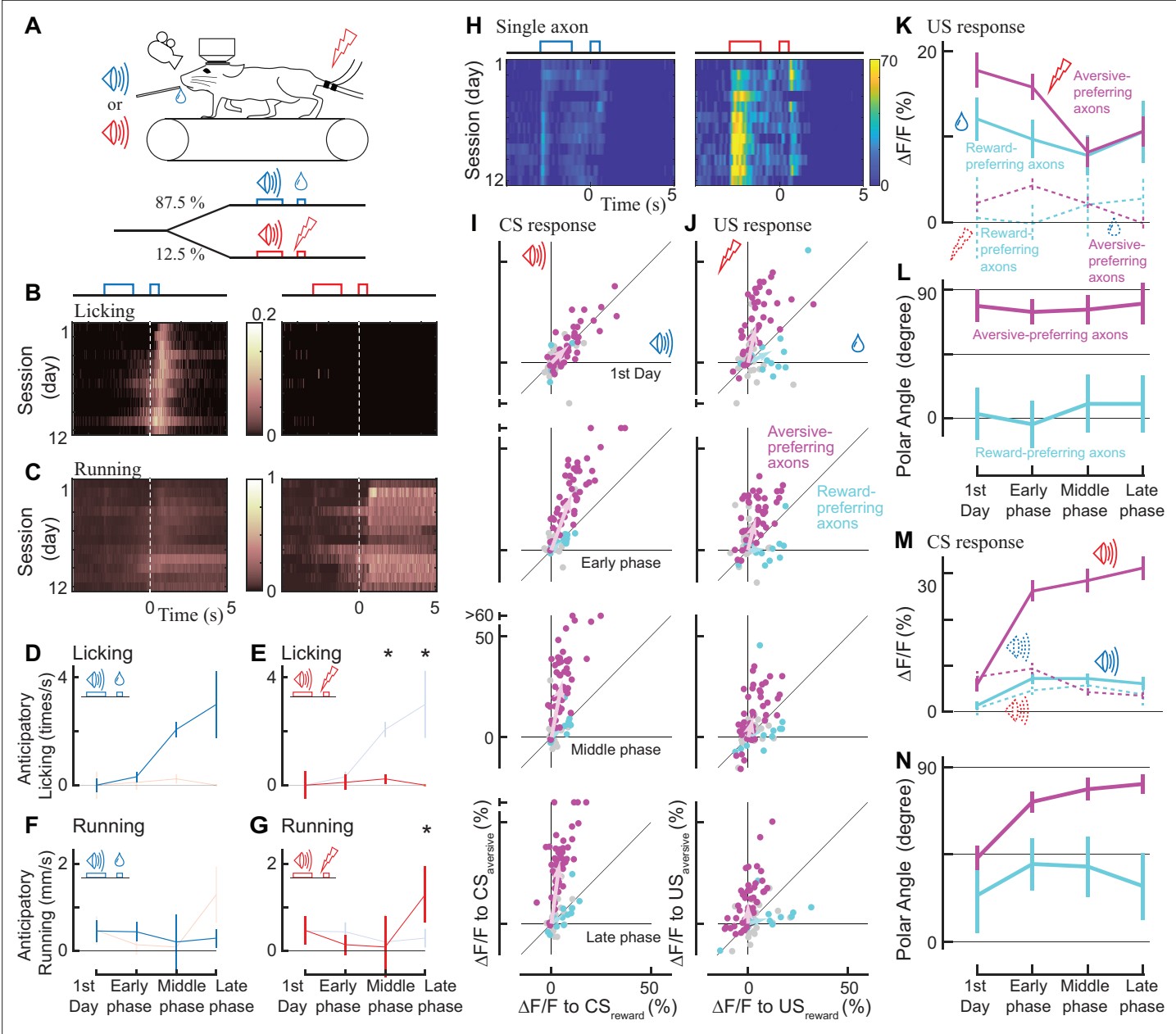

**Figure 3.** Classical conditioning induced behavioral and neural changes. (**A**) Experimental design. Auditory cues were presented before unconditioned stimuli (rewarding or aversive stimuli). (**B, C**) Behavioral changes in one example animal across 12 days (B: licking, C: running). In the reward condition, the animal gradually developed anticipatory licking (B, left). In the aversive condition, the animal usually ran after the shock delivery, but sometimes even before the delivery (C, right). Licking and running traces were normalized to the instantaneous maximum values for this animal and then averaged over a single day. (**D–G**) Anticipatory behavior during classical learning (n=6 animals). In the late phase of learning, anticipatory licking was primarily observed in the reward condition (**D**) but not in the aversive condition (**E**). Anticipatory running was seen more often in the aversive condition (**G**) than in the reward condition (**F**). (**H**) Activity change in one example axon across 12 days. The axon was from the same animal shown in B, C. (**I, J**) Learning induced changes in response to conditioned cues (**I**) and unconditioned stimuli (**J**) for aversive-preferring axons (magenta) and reward-preferring axons (cyan) together with non-significant axons (black). Aversive- and reward-preferring axons were defined before the start of classical training. The x-axis represents the reward condition, and the y-axis represents the aversive condition. n=47 for aversive-preferring axons, n=12 for reward-preferring axons. Vector averages representing aversive-preferring axons and reward-preferring axons are depicted as magenta and cyan arrows, overlaid in each panel. (**K**) Learning induced a change in the amplitude of unconditioned response of aversive-preferring axons for the aversive condition (magenta solid line) and reward condition (magenta dotted line) and that of reward-preferring axons for reward condition (cyan solid line) and aversive condition (cyan dotted line). Number of axons is the same as I, J. (**L**) The polar angle of the scatter plot in J. The magenta line represents aversive-preferring axons and the cyan line represents reward-preferring axons. (**M**) Similar to K, but for the conditioned response. (**N**) Similar to L, but for the conditioned response.

*Figure 3 continued on next page*

*Figure 3 continued*

The online version of this article includes the following figure supplement(s) for figure 3:

**Figure supplement 1.** Anticipatory licking and running during the classical conditioning (n = 6).

**Figure supplement 2.** Population activity of aversive- and reward-preferring axons throughout learning (aversive-preferring axons, n = 47; reward-preferring axons, n =12).

**Figure supplement 3.** Activity is not suppressed during reward omission.

**Figure supplement 4.** Response in the first trials in each session.

magenta, p=0.002 for rewarding stimuli, p=0.007 for aversive stimuli, n=47; Wilcoxon signed-rank test, comparison between the first day and the last phase), maintaining similar preferences for rewarding and aversive stimuli (*Figure 3L*, magenta, p=0.24, n=47; circular statistics, comparison between the first day and the last phase). Similarly, reward-preferring axons maintained their preferences over the course of classical conditioning (*Figure 3L*, cyan, p=0.77, n=12; circular statistics, *Figure 2—figure supplement 2B*).

Next, we quantified the activity change at the time of the conditioned auditory cues (CS$_{reward}$ and CS$_{aversive}$, *Figure 3I*) throughout the learning process. On the first day, aversive-preferring axons already showed a transient response to conditioned cues, implying that the conditioned stimulus response was not acquired through learning (first day in *Figure 3H and M*, and *Figure 3—figure supplement 2*). In addition, the conditioned stimulus response showed no particular reward/aversive preference (*Figure 3N*, first day, for aversive-preferring axons, p=0.66, n=47), indicating that aversive-preferring axons did not distinguish the two conditioned cues. However, at the later phases of learning, the conditioned stimulus response was enhanced for CS$_{aversive}$ in aversive-preferring axons (*Figure 3M*, magenta, p<0.0001, n=47; Wilcoxon signed-rank test, comparison between the first day and the last phase) and slightly attenuated for CS$_{reward}$ (p<0.001), resulting in a stronger preference for aversive processing (late phase in *Figure 3N*, magenta, p<0.001). In contrast, for reward-preferring axons, the conditioned stimulus response increased both for CS$_{aversive}$ (*Figure 3M*, dotted cyan line, non-significantly, p=0.09, n=12, Wilcoxon signed-rank test) and for CS$_{reward}$ (solid cyan line, significantly, p<0.007), resulting in an unchanged preference (*Figure 3N*, cyan, p=0.77). Consistently, these changes in CS and US responses across the training phases were evident in the population-averaged calcium response (*Figure 3—figure supplement 2*).

We also tested whether the dopamine axons showed suppressed activity when the predicted reward was omitted, one of the major features of reward prediction error coding (*Schultz et al., 1997*; *Engelhard et al., 2019*; *Amo et al., 2022*). Such activity suppression has been detected with GCaMP6m at cell bodies of dopamine neurons (*Engelhard et al., 2019*) as reduced signal at 0–4 s after the delivery of reward. Therefore, we included one condition for an unexpected reward omission on the last day of the late phase of the classical conditioning (*Figure 3—figure supplement 3*). We found that upon the reward omission, the reward-preferring dopamine axons did not show activity suppression, indicating that the mPFC dopamine axons do not respond to reward omission.

Taken together, our two-photon imaging revealed that a minority of mPFC dopamine axons prefer reward activity (reward-preferring axons), and that these axons are not involved in reward prediction error in a classical learning paradigm. In contrast, the majority of dopamine axons are strongly involved in aversive-related processing (aversive-preferring axons), and the preference for the aversive cue is enhanced through classical conditioning.

## Dopamine axons show enhanced selectivity of cue activity in trials with correct discrimination

In the classical conditioning paradigm, an enhanced preference of aversive-preferring dopamine axons for aversive cues (*Figure 3N*) was accompanied by improved behavioral discrimination of the two conditioned cues (*Figure 3D–G*). Based on this finding, can correct cue discrimination be accompanied by an enhanced neural preference when animals make trial-by-trial judgements in discriminating cues even after conditioning?

To investigate trial-by-trial judgements of conditioned cues, we classified the trials into four groups (*Figure 4A*) based on correct or incorrect discriminating behavior. First, we focused on the presence or absence of anticipatory licking, as the licking behavior can discriminate the two

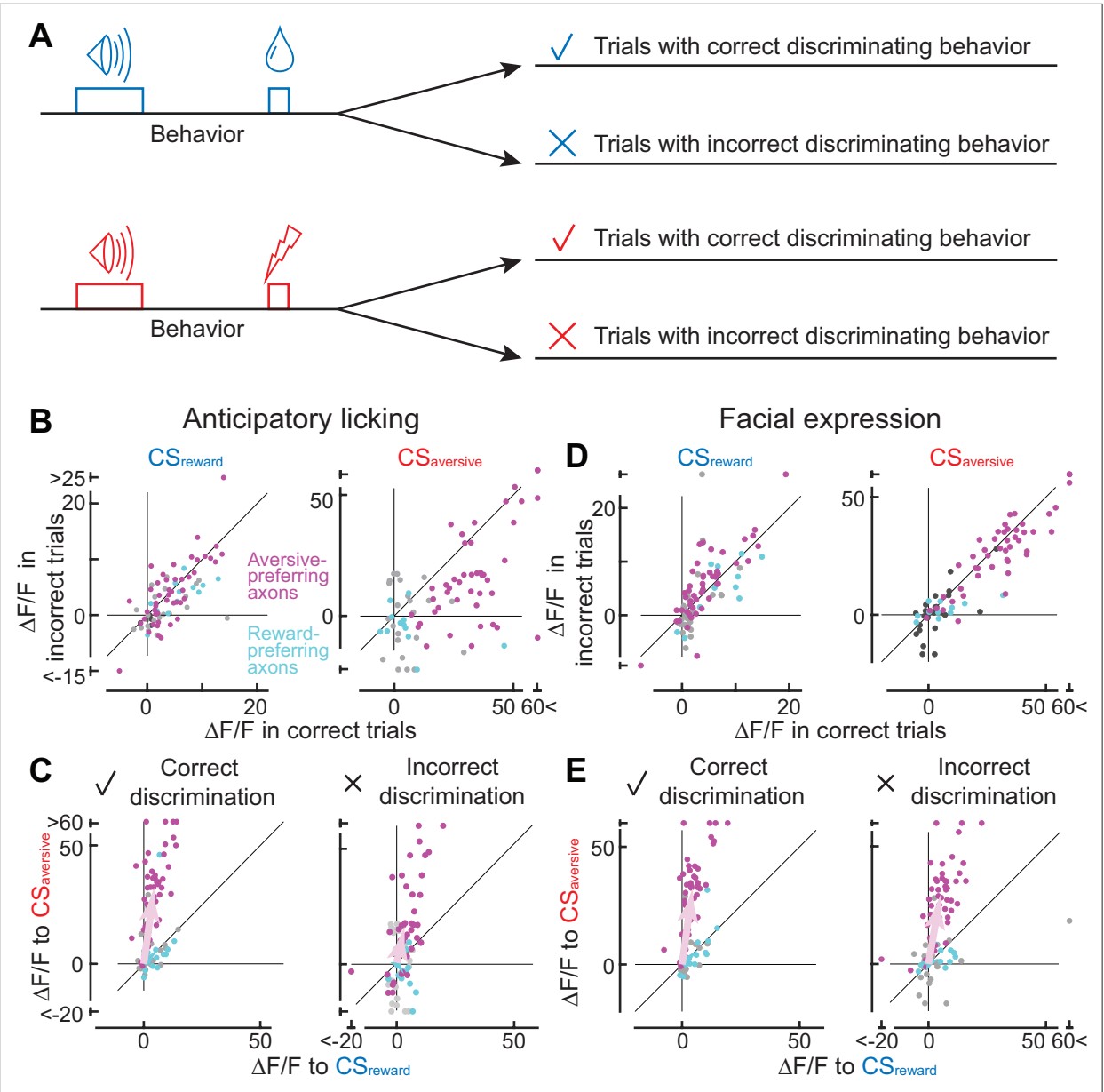

**Figure 4.** Axonal cue response in trials with correct or incorrect cue discrimination. (**A**) Classification of trials based on the behavioral response that occurred between the conditioned stimulus onset and unconditioned stimulus onset. Such behaviors include anticipatory licking or facial expressions. (**B**) Comparison of cue response between correct (x-axis) and incorrect (y-axis) trials based on anticipatory licking (magenta, aversive-preferring axons, n=44; cyan, reward-preferring axons, n=12). The left panel shows the reward cue response, and the right panel shows the aversive cue response. (**C**) Preference for reward or aversive predictive cues in correct (left panel) and incorrect (right) discrimination. Magenta represents aversive-preferring axons (n=44) and cyan represents reward-preferring axons (n=12). Vector averages representing aversive-preferring axons and reward-preferring axons are depicted as magenta and cyan arrows. Note stronger preference for aversive cue as a population in correct discrimination. (**D**) Similar to B, but based on facial expressions (magenta, aversive-preferring axons, n=47; cyan, reward-preferring axons, n=12) (**E**) Similar to C, but based on facial expressions (magenta, aversive-preferring axons, n=47; cyan, reward-preferring axons, n=12). As in C, preference for the aversive cue is stronger in correct discrimination.

The online version of this article includes the following figure supplement(s) for figure 4:

**Figure supplement 1.** Machine learning discriminates auditory cues based on anticipatory licking.

**Figure supplement 2.** Discrimination based on facial expressions.

conditioned stimulus tones, particularly at the late phase of learning (*Figure 4—figure supplement 1*, based on the random forest classifier). The first group exhibited licking after CS$_{reward}$ (correct reward discrimination), the second group exhibited no licking after CS$_{reward}$ (incorrect reward discrimination), the third group displayed no licking after CS$_{aversive}$ (correct aversive discrimination), and the fourth group displayed licking after CS$_{aversive}$ (incorrect aversive discrimination). The classification is invalid when animals make random guesses (discrimination of 50%), so we focused on results from the late phase of learning (or the middle phase if there were no errors in anticipatory licking in the late phase).

Can the axonal response to conditioned cues be impacted by whether animals discriminate the cues correctly or incorrectly? An incorrect discrimination of the aversive cue is accompanied by the presence of anticipatory licking, resulting in error trials in our machine learning-based analysis. Such error trials (*Figure 4A*, fourth group) occurred in 1.6% of cases, showing a weaker aversive cue response than correct trials (third group; p<0.0001, n=44 magenta axons, Wilcoxon signed-rank test, *Figure 4B*, right). In contrast, the absence of anticipatory licking despite the reward-predictive cue comprises another type of error (second group, 49.0%). In such error trials, the reward cue response was not significantly different from that in the correct trials (first group; p=0.26 n=44 axons, Wilcoxon signed-rank test, *Figure 4B*, left). Overall, the reward/aversive preference was stronger in correct discrimination trials than in incorrect trials (left vs. right in *Figure 4C*, magenta, 82.2°±1.1° vs. 70.0°±7.6°, p=0.049, circular statistics).

In addition to anticipatory licking, the discrimination of predictive cues can be inferred by the facial expressions of mice. Facial expressions of mice can capture emotional states (*Dolensek et al., 2020*), and have been used to make binary judgements of the presence or absence of pain with the application of a deep neural network (*Tuttle et al., 2018*). In this study, we combined a pretrained deep neural network (ResNet3D) (*Tran et al., 2018*) and a machine learning classifier (random forest classifier; *Breiman, 2001*) to make binary judgements of whether the animals experienced reward or aversive conditions, based on facial expressions during the cue presentation (*Figure 4—figure supplement 2*). The percentage of errors in discrimination during the 2 s cue presentation was 16.0% ± 1.4% (n=6 animals), comparable to the result for anticipatory licking during the cue plus delay periods (25.3% ± 6.1%). However, discrimination based on facial expressions resulted in a higher number of error trials in aversive conditions than discrimination based on licking (14.1% vs 1.6%), and a lower number in reward conditions (18.0% vs 49.0%). This discrepancy might be explained either by temporal discrepancy between the cue period (facial expression) and the delay period (most cases of anticipatory licking) or by the fact that anticipatory licking might represent reward uncertainty rather than reward expectation (*Ogawa et al., 2013*).

Correct cue-discrimination based on facial expressions analysis also revealed sharper selectivity for reward or aversive cues. A trial-by-trial error analysis revealed that the axonal activity to CS$_{aversive}$ was enhanced in the correct trials in the aversive-preferring axons (*Figure 4D*, right, p=0.023, n=47 axons, Wilcoxon signed-rank test), consistent with the analysis based on anticipatory licking (*Figure 4B*, right). In addition, the response to CS$_{reward}$ was significantly weaker in the correct trials (*Figure 4D*, left, p=0.019, n=47 axons, Wilcoxon signed-rank test). As a result, the reward/aversive preference was stronger in correct discrimination trials than in incorrect trials for the aversive-preferring axons (left vs. right in *Figure 4E*, magenta, 84.7°±3.0° vs. 81.7°±4.0°, p=0.019, circular statistics).

In contrast to the aversive-preferring axons, correct discrimination had no effect on the CS activity of the reward-preferring axons. We found that the response to CS$_{reward}$ and CS$_{aversive}$ was not significantly different between correct and incorrect judgement trials (cyan points in *Figure 4B and D*, anticipatory licking: CS$_{aversive}$ p=0.052, facial expressions: CS$_{reward}$ p=0.08, CS$_{aversive}$ p=0.20, n=12, Wilcoxon signed-rank test) except for the CS$_{reward}$ response based on anticipatory licking (p=0.016, n=12). As a result, selectivity for CS$_{reward}$/CS$_{aversive}$ was not improved in correct trials (anticipatory licking: p=0.15, facial expressions: p=0.39, cyan points in *Figure 4C and E*). Therefore, correct/incorrect discrimination impacts aversive- and reward-preferring axons differentially.

Altogether, when animals exhibited the correct behavioral response (either anticipatory licking or facial expression), aversive-preferring but not reward-preferring axons showed a higher selectivity for aversive cue processing (*Figure 4C and E*).

## Discussion

Dopamine projections to the mPFC are considered one of the key neuromodulators that enable flexibility in neural processing of the mPFC. However, due to technical difficulties in recording the dopamine neurons of specific projections, little is known about the signals conveyed by mPFC projections, including the basic question of whether individual projections signal reward or aversive information. In this study, we optimized a two-photon imaging approach based on a microprism to image the calcium activity of dopaminergic axons in the mPFC. We uncovered differences in reward/aversive preferences in individual dopamine axons with an overall preference for aversive stimuli. In addition, we demonstrated that aversive-preferring axons responded equally to reward- and aversive-predictive conditioned cues in classical conditioning on the first day; however, this response became strongly biased toward the aversive conditioned cue through the conditioning. Finally, based on a trial-by-trial analysis of the animals' behavior following reward- or aversive-predictive cues, we found that aversive-preferring axons exhibited higher selectivity for cues when the cues were successfully discriminated behaviorally.

Our study revealed functional diversity in mPFC-projecting dopamine axons by addressing a long-standing question of whether dopamine neurons send reward- or aversion-related signals to the mPFC (*Weele et al., 2019*; *Verharen et al., 2020*). The activity of mPFC-projecting dopamine neurons can be investigated extracellularly by incorporating antidromic stimulation (*Mantz et al., 1989*), but this approach is laborious. Thus, many studies have used more technically feasible but less direct approaches, particularly for awake animals, such as measuring dopamine release with microdialysis (*Abercrombie et al., 1989*; *Bassareo et al., 2002*), measuring catecholamine release with fast-scan cyclic voltammetry recently combined with optogenetic and pharmacological identification (*Vander Weele et al., 2018*), and measuring bulk calcium activity from dopamine axons with fiber photometry (*Kim et al., 2016*; *Ellwood et al., 2017*). These studies have led to somewhat inconsistent conclusions: some studies have reported reward signals whereas others have reported aversive signals. Reconciling these findings is challenging, as different studies have used different approaches to assess the effects of either a rewarding or an aversive stimulus, but not both. Our two-photon imaging approach provided a unique opportunity to compare rewarding and aversive signals of individual projection neurons (i.e. individual axon projections). Our comparison revealed diversity in the dopamine axons, and that many dopamine axons responded to both rewarding and aversive stimuli, with a strong bias for aversive stimuli at the population level. However, this population bias was not fixed; rather, the bias depended on both the reward volume and the intensity of the aversive stimulus. In addition, the bias might also rely on of the frequency of rewarding and aversive events, which our study could not address; we presented aversive stimuli less frequently to keep mice engaged. All these parameters may collectively explain why some studies have reported a strong response to aversive stimuli but little response to rewards. Our study revealed functional diversity in dopamine axons in the superficial layers, but did not address whether this diversity could also be found in axons in the deep layers. It is not clear whether single axons have branches in both the superficial and deep layers, even in anatomical studies that used single-cell tracing of dopamine neurons (*Matsuda et al., 2009*; *Aransay et al., 2015*). Further investigation into these layers may reveal a richer functional diversity in dopamine axons in the mPFC.

The firing of mPFC-projecting dopamine neurons cannot be simply explained by value coding performed by conventional midbrain dopamine neurons; their firing rates increase in response to opposite hedonic valences, rewarding and aversive stimuli, exhibiting diverse preference. Their firing might be captured by salience coding, which includes motivational salience signals and alerting signals (*Bromberg-Martin et al., 2010*). Similarly, salience coding may explain the response of aversive-preferring axons to aversive-predictive cues. On the first day of classical conditioning, when the animals had not yet established a link between conditioned and unconditioned stimuli, the aversive-preferring axons showed a transient activity increase to two types of conditioned cues with no bias, implying that activity serves as an alerting signal (*Vander Weele et al., 2018*). After days of training, the activity became strongly biased toward the aversive cue, indicating that the activity might additionally encode a motivational salience signal (*Bromberg-Martin et al., 2010*; *Lee et al., 2021*). In our study, motivational salience might appear to play a significant role in the processing of the aversive predictive cue. Specifically, in aversive-preferring axons, the activity associated with the aversive cue exhibits distinct trends within daily sessions compared to those of the reward cue and unconditioned

cue activities (magenta points in *Figure 3—figure supplement 4*). The response of the aversive-preferring axons may be useful for the recipient mPFC to allocate its resources to the most salient outcomes and their predictors, a proposed role of the mPFC (*Desimone and Duncan, 1995*; *Miller and Cohen, 2001*; *Ridderinkhof et al., 2004*; *Isoda and Hikosaka, 2007*; *Nee et al., 2011*; *Bissonette et al., 2013*; *Sharpe et al., 2019*). The response of reward-preferring axons also does not follow value-coding, considering the absence of reward omission suppression. It is an open question whether saliency coding plays more important roles in aversive processing than in reward processing. To clarify the detailed nature of the saliency coding, together with the functional diversity of mPFC dopamine axons, further study is necessary. Such a study should include different types of unconditioned stimuli, systematically vary the physical features of conditioned and unconditioned stimuli, and separate the motivational salience and alerting signals with different task designs.

Consistent with salience coding without hedonic valences, phasic optogenetic stimulation of dopamine axons in the mPFC does not reinforce or suppress any behavioral actions (*Popescu et al., 2016*; *Ellwood et al., 2017*; *Vander Weele et al., 2018*) (but see [*Gunaydin et al., 2014*]). Instead, optogenetic stimulation can increase the signal-to-noise ratio of aversive processing in mPFC neurons for competitive situations in which reward and appetitive cues are simultaneously presented (*Vander Weele et al., 2018*). Our results are consistent with the recent view that dopamine at the mPFC gates sensory inputs for aversive processing (*Ott and Nieder, 2019*; *Weele et al., 2019*).

Using aversive classical conditioning, we revealed that aversive learning can induce activity changes in dopamine axons. The classical conditioning is a form of aversive learning distinct from instrumental aversive learning including punishment and active avoidance (*Jean-Richard-Dit-Bressel et al., 2018*). Although all types of aversive learning are processed in the mPFC and dopamine systems, each type may be expected to include distinct neural circuits. Further research is necessary to reveal the detailed processing of aversive learning in the mPFC and dopamine projections.

Our study provides new insights into the functional diversity of dopamine neurons that constitute mesocortical pathways. Previous studies employing fiber photometry imaging have identified functional diversity among dopamine neurons with distinct projection pathways; dopamine axons in the ventral nucleus accumbens medial shell (*de Jong et al., 2019*; *Yuan et al., 2019*), in the tail of the striatum (*Menegas et al., 2017*), and in the basal amygdala (*Lutas et al., 2019*) do not show activity that matches reward prediction error coding, but instead show increased activity for aversive stimuli (*Verharen et al., 2020*). Our two-photon imaging results demonstrate that even the same projection-defined dopamine neurons can be inhomogeneous, with some preferring aversive signals and others preferring reward signals. The aversive response found in some dopamine pathways, including mesocortical dopamine projections, might be linked to glutamate co-release, as vesicular glutamate transporter 2 (*Slc17a6*) genes are expressed in dopamine neurons projecting to the ventral nucleus accumbens medial shell, the tail of the striatum, and the mPFC (*Poulin et al., 2018*), and as AMPA-receptor-mediated excitatory postsynaptic currents have been confirmed upon the stimulation of dopamine axon terminals in the basal amygdala (*Lutas et al., 2019*). These glutamate co-releasing dopamine axons might even be collaterals of the same dopamine neurons; a single-cell tracing study showed examples of dopamine neurons with their axon collaterals in the PFC and the basal amygdala or those in the PFC and nucleus accumbens shell (*Aransay et al., 2015*). Meanwhile, dopamine axons projecting to the PFC are not just from *Slc17a6*-expressing neurons but also from *Slc17a6*-negative neurons (*Poulin et al., 2018*). One possible scenario is that *Slc17a6* may be expressed in aversive-preferring axons in the mPFC but not in reward-preferring axons. This scenario might be in line with a recent fibermetry study on genetic features of dopamine neurons in the SNc, where $Slc17a6^+$ dopamine neurons showed a strong aversive and weak reward response as a population, whereas $Slc17a6^-/Calb1^+$ dopamine neurons showed a weak aversive and strong reward response (*Azcorra et al., 2023*). As of now, it is not clear how functional diversity within the same mesocortical pathway is linked to molecular diversity. Clarifying such a link will further advance our understanding of distinct dopamine subsystems and may shed light on how dopamine subsystems are dysregulated in prefrontal psychiatric diseases.

## Materials and methods

**Key resources table**

| Reagent type (species) or resource | Designation | Source or reference | Identifiers | Additional information |
|---|---|---|---|---|
| Antibody | Anti-GFP (Rabbit Polyclonal) | Thermo Fisher Scientific | Cat# A-11122, RRID:AB_221569 | IF (1:1000) |
| Antibody | Anti-Tyrosine Hydroxylase (Sheep Polyclonal) | Abcam | Cat# ab113, RRID:AB_297905 | IF (1:200) |
| Antibody | Anti-Rabbit IgG (H+L) Antibody, Alexa Fluor 488 Conjugated (Donkey Polyclonal) | Thermo Fisher Scientific | Cat# A-21206, RRID:AB_2535792 | IF (1:500) |
| Antibody | Anti-Sheep IgG (H+L) Antibody, Alexa Fluor 568 Conjugated (Donkey Polyclonal) | Thermo Fisher Scientific | Cat# A-21099, RRID:AB_2535753 | IF (1:500) |
| Recombinant DNA reagent | hSynapsin1-FLEx-axon-jGCaMP8m (plasmid) | This paper | Addgene #216533 | Described at 'Headplate implant and virus injection' of 'Surgery' section |
| Strain, strain background (AAV) | AAV2/1-hSynapsin1-FLEx-axon-jGCaMP8m-WPRE-SV40 | University of South Carolina Viral Vectors Core | | |
| Genetic reagent (mouse) | Mouse: Slc6a3$^{tm1.1(cre)Bkmn}$ | The Jackson Laboratory | JAX: 006660 | |
| Chemical compound | Normal Donkey Serum | Sigma-Aldrich | Cat# D9663, RRID:AB_2810235 | |
| Software, algorithm | MATLAB | Mathworks | RRID: SCR_001622 https://www.mathworks.com | |
| Software, algorithm | Python | Mathworks | RRID:SCR_008394 https://www.anaconda.com/ | |
| Software, algorithm | Suite2p | *Patriarchi et al., 2018*; *MouseLand, 2024* | https://github.com/MouseLand/suite2p | |
| Software, algorithm | DeepLabCut | *Mathis et al., 2018* | https://github.com/DeepLabCut/DeepLabCut; *Mathis et al., 2024* | |
| Software, algorithm | B-spline Grid, Image and Point based Registration | Dirk-Jan Kroon | https://jp.mathworks.com/matlabcentral/fileexchange/20057-b-spline-grid-image-and-point-based-registration | |
| Software, algorithm | Pytorch, TorchVision | Meta AI | https://pytorch.org/ | |
| Other | Mouse Brain Connectivity Atlas | Allen Brain Map | https://connectivity.brain-map.org/ | Further details are provided in the caption of *Figure 1—figure supplement 1*. |

## Experimental model details

### Animals

All experimental procedures were approved by local institutions supervising animal experiments at the Medical University of South Carolina, Monash University, Kagoshima University. Heterozygous dopamine transporter (DAT)-Cre mice (Slc6a3$^{tm1.1(cre)Bkmn}$, Jackson Laboratory, #006660, crossed with wild-type C57BL/6) was used in this study, including 12 mice for two-photon imaging and 10 for histology. Previous research utilized the same mouse line to express GCaMP6f in dopamine axon terminals in the mPFC that could be detected by one-photon fiber photometry (*Kim et al., 2016*). Mice of both sexes, aged >8 weeks were included. The mice were maintained in group housing (up to five mice per cage) and experiments were performed during the dark period of a 12 hr light/12 hr dark cycle.

## Method details

### Surgery

All surgical procedures were performed aseptically, with the mice under anesthesia with isoflurane. Lidocaine (subcutaneously at the incision), atropine (0.3 mg/kg, intraperitoneally), caprofen (5 mg/kg, intraperitoneally), and dexamethasone (2 mg/kg, intraperitoneally) were applied to prevent pain and

brain edema. After surgery, the mice were allowed to recover for at least three days. No experimenter blinding was done.

### Headplate implant and virus injection

A custom-made headpost was glued and cemented to the skull, and then, a small craniotomy (<0.5 mm) was performed over the VTA (~2.9–3.5 mm posterior and ~0.5 mm lateral from the bregma). Inside the small craniotomy, axon-GCaMP virus (AAV2/1-hSynapsin1-FLEx-axon-jGCaMP8m-WPRE-SV40) was volume-injected (Nanoject III, Drummond Scientific) to the VTA through a pulled capillary glass (40–60 nL/site; depth: 4200–4400 μm; 15 min/injection). After the injection, the craniotomy was sealed with a small piece of cover glass and silicon sealant (Kwik-Cast) and animals were returned to their home cage.

For axon-GcaMP, we synthesized an axon-jGCaMP8m construct based on GAP43 (*Broussard et al., 2018*), a linker (*Broussard et al., 2018*), and jGCaMP8m (*Zhang et al., 2023*), together with restriction sites for SpeI and AscI. Then, the construct was inserted into a hSynapsin1-FLEx vector, to make hSynapsin1-FLEx-axon-jGCaMP8m-WPRE-SV40. The plasmid has been deposited into Addgene (#216533).

### Microprism implant

After a 3-week waiting period of adeno-associated virus (AAV) expression, a microprism was inserted for two-photon imaging as described previously (*Low et al., 2014*). A rectangular craniotomy (4x2 mm) was made over the bilateral PFC (~1.5–3.5 mm anterior from the bregma), and the dura was removed over the right hemisphere. Then, a microprism implant assembly was inserted into the subdural space within the fissure (*Figure 1B, E and F*). The microprism was centered ~2.5 mm anterior to the bregma to avoid damaging bridging veins. Once implanted, the prism sat flush against the opposing fissure wall, which contained the medial wall of the PFC (mainly the prelimbic area) in the left hemisphere. The front face of the prism was oriented along the midline.

The assembly consisted of a right-angle microprism (2x2 x 1 mm, Prism RA N-BK7, Tower Optical Corp.) and two coverslip layers (top layer: 4.5x3.0 mm, bottom layer: 3.6x1.8 mm), which were glued by ultraviolet curing optical adhesive (Norland #81). The top layer of glass was cemented to the skull with dental acrylic. Our assembly design (microprism of 2x2 x 1 mm, plus double-layer glass) is different from the original report (microprism of 1.5x1.5 x 1.5 mm plus single-layer glass; *Low et al., 2014*) for the following reasons. First, the thinner microprism (1 mm in the anterior-posterior axis) was easier to insert into the bank, avoiding superficial veins branching from the superior sagittal sinus. Second, the longer prism (2 mm in the dorsal-ventral axis) could spare a wider imageable region below the superior sagittal sinus. Third, the double-layer glass helped suppress brain movements.

## Behavior

After the microprism implant surgery, the mice were allowed to recover in their home cages for one week. After recovery, the mice underwent water scheduling (receiving 0.8–1 mL of water per day). Then, the mice were pretrained for head fixation and for drinking water from a spout on a linear passive treadmill (SpeedBelt, Phenosys) in a sound-proof blackout box for two days. After the initial days of reward only, the animals received infrequent electrical shocks interspersed with the reward. Once the animals experienced both reward and shock conditions, we started the two-photon imaging sessions.

To monitor licking behavior, the face of each mouse was filmed with a camera at 65 Hz (CM3-U3-13Y3M-CS, FLIR) using infrared illumination (850 nm light-emitting diode, IR30, CMVision or M850F2, Thorlabs). To detect locomotion, the running speed on the treadmill was recorded at 30 kHz.

### Rewarding and aversive stimuli

The mice received rewarding or aversive stimuli with unpredictable timing. The stimuli were administered in a randomized order (rewarding stimuli: seven out of nine cases; aversive stimuli: one out of nine; control period: one out of nine), with a randomized inter-trial interval of 55–65 s. The mice exhibited comfortable behavior on the treadmill for 1.5–2 hr.

As a reward, 10 μL of sugar water was delivered through a water spout (*Figure 2A*), controlled by a TTL pulse (200ms) delivered to a syringe pump (PHM-107, Med Associates, Inc, USA). Based on

previous literature, a 10 µL reward is relatively large (*Tsutsui-Kimura et al., 2020*). Animals typically underwent 100–200 reward trails. In some experiments, the reward volume was varied between 0 and 15 µL (*Figure 2—figure supplement 4*). As an aversive stimulus (*Kim et al., 2016*; *de Jong et al., 2019*; *Lutas et al., 2019*), a 1 s, 0.2-mA electrical current was delivered via a stimulator (AM2100, A-M systems, USA) between two electrode pads attached to the mouse's tail (*Figure 2A*). This current was considered to be mild, just strong enough to evoke locomotion. When the current was doubled (*Figure 2—figure supplement 4*), the locomotion tended to become stronger, but some animals stopped drinking water. Similarly, when the frequency of the aversive stimuli was increased (e.g. 50% of trials), some mice were no longer motivated to drink the reward water.

## Classical conditioning

After three days of reward and aversive stimulus sessions, we trained the mice in reward and aversive trace conditioning. The structure of the task is the same as that for the reward and aversive stimulus sessions (reward condition: seven out of nine cases; aversive condition: one out of nine; control condition: one out of nine; inter-trial interval: 55–65 s), except that auditory stimuli (9 or 13 kHz, 2 s) were presented 3 s before presenting the unconditioned stimuli (rewarding or aversive stimulus). Anticipatory behavioral responses confirmed that the mice could discriminate the tone frequency differences (*Figure 3D–G*), consistent with a previous report showing that mice can discriminate frequency differences down to 4–7% (*de Hoz and Nelken, 2014*). In three animals, 9 kHz tone was used for the reward-predictive cue, 13 kHz for the aversive-predictive cue. In the remaining three animals, the tone association was reversed.

We separated the learning into three phases in addition to the first day. In the late phase, anticipatory licking and running reach a saturation level, as evidenced by learning curves spanning six periods (*Figure 3—figure supplement 1*).

When the anticipatory licking was stably manifested (*Figure 3D*, late phase), we included one condition for an unexpected reward omission among the seven reward conditions (*Figure 3—figure supplement 3*) and continued for two more days.

## Timing control

Synchronization of the two-photon imaging, behavior camera image acquisition, reward delivery, aversive stimulation, and sound presentation were achieved using digital and analogue output from a National Instruments board (NI USB-6229), which was controlled by a custom-made MATLAB program. We also recorded continuous signals (sampled at 30 kHz, PCIe-6363, National Instruments) of the treadmill speed, the frame timing of two-photon imaging, the frame timing of a behavior camera, copies of command waves to the syringe pump, the stimulator, and the speaker.

## Two-photon imaging

In vivo two photon imaging was performed using a table-mounted microscope (Bergamo II, Thorlabs or MOM, Sutter Instruments) and a data acquisition system. The light source was a pulsed Ti:sapphire laser (MaiTai DeepSee eHP, SpectraPhysics, or Chameleon Ultra II, Coherent) with dispersion compensation, with the laser wavelength set to 980 nm (*Hasegawa et al., 2017*; *Itokazu et al., 2018*), which causes a higher fluorescent change in the GCaMP signal and less scattering in the tissue than 920 nm. The laser power at the apochromatic objective lens (16×, 0.80 NA, Nikon) was <70 mW, and we saw no bleaching. Green fluorescent photons were filtered (ET525/70 m-2p) and collected by a hybrid photodetector (R11322U-40–01, Hamamatsu Photonics) (*Tischbirek et al., 2015*) and a high-speed current amplifier (DHPCA-100, Femto). Imaging frames were acquired at ~60 Hz and were downsampled offline. Images were collected at a depth of 30–100 µm from the dural surface (up to ~200 x 200 µm). The small field of view at a high sampling rate makes it possible to collect weak signals from small structures, as in spine functional imaging (*Jia et al., 2014*).

Imaging fields were searched based on the presence of fiber morphology with at least occasional calcium transients in the fibers, not based on the behavioral correlation of the transients. Fiber morphology and spontaneous calcium transients were not reliably visible in axons deeper than 100 µm in a live-view mode, possibly because of the low signal-to-noise ratio. For each mouse, imaging was performed for a single field per day in order to gain a sufficient number of repeats with a 1 min inter-trial interval. In reward/aversive preference characterization (*Figure 2*), 1–2 sites were imaged

on different days. For the classical conditioning, only a single site was imaged during the course of conditioning. Once the imaging site was determined on the first day, the reference image of two-photon imaging was captured, in addition to the surface vessel image of one-photon imaging. On subsequent days, these images were used to return to the same imaging site and depth, comparing and overlaying the reference image and the ongoing imaging view.

## Calcium imaging data analysis

### Data processing

Imaging data was processed for motion correction and registration. Axons were detected for region-of-interest (ROI) drawing using Suite2p (*Pachitariu et al., 2016*) and a custom-made MATLAB program (*Itokazu et al., 2018*). A fluorescent trace for each ROI was generated, and then the trace was normalized by the baseline fluorescence (F0, set as the 50th percentile fluorescence over a 30 s sliding window in order to remove any slow drifts in the baseline) to produce a ΔF/F trace.

Dopamine axons were sparsely labeled in the mPFC, but the same axons needed to be excluded based on correlation analysis among pairs (*Petreanu et al., 2012*; *Sun et al., 2016*; *Itokazu et al., 2018*). The correlation coefficients of ΔF/F traces were calculated for axons in each plane, and pairs showing a higher correlation (>0.65; *Itokazu et al., 2018*) were considered to arise from the same axon. The high correlation pairs were grouped into clusters, and the ROI with the largest ΔF/F signal in each cluster was assigned to represent the cluster. The aforementioned procedure was iterated repeatedly until the correlation between all remaining pairs fell below the threshold of 0.65. Our results remained similar for different correlation threshold.

### Reward, aversive, cue, and locomotion activity

For each axon, reward and aversive activity were evaluated. Reward activity was quantified as an increase in ΔF/F by comparing the average ΔF/F between the control range (−2–0 s from the onset of the reward TTL to a syringe pump) and the signal range (0–2 s). Similarly, aversive activity was quantified as an increase in ΔF/F, based on the difference between the average ΔF/F between the control range (−2–0 s from the onset of the electrical shock TTL) and the signal range (0–2 s). Axons were considered to exhibit a significant response if the magnitude of either activity was statistically larger than that of the baseline activity (Wilcoxon signed-rank test; $p < 0.05$). Significant axons were classified as either reward-preferring (cyan) or aversive-preferring clusters (magenta) based on k-means clustering, the separation of which coincided approximately with the unity line of the reward/aversive scatter plot, as shown in *Figure 2J*.

The locomotion activity was quantified as an increase in ΔF/F by comparing the average ΔF/F between the control range (−2–0 s from the locomotion initiation) and the signal range (0–2 s). The locomotion initiation is defined in the 'Running detection' section below.

During classical conditioning, activity was evaluated in a similar manner. For the conditioned cue activity, the activity increase was computed by comparing the average ΔF/F between the control range (−2–0 s from the onset of the predictive cue) and the signal range (0–2 s from the cue onset). For the unconditioned response activity (reward or aversive), we compared the control range (−2–0 s from the onset of the predictive cue) and the signal range (0–2 s from the onset of the unconditioned stimulus). To investigate the preference for reward or aversive processing, we used scatter plots (*Figure 3I and J*), similar to *Figure 2J*. The color-coded classification (cyan/magenta) was based on k-means clustering, using the responses before classical conditioning (*Figure 2J*).

### Evaluation of brain movement

To compare the amount of brain movement between the two different microprism assemblies (*Figure 1—figure supplement 2*), we obtained x- and y-axis shifts of acquired images caused by the brain movement. The shifts were computed by the Suite2p program and used for image registration (*Pachitariu et al., 2016*). We quantified the brain shift using two metrics: root mean square and large transient movement. First, the root mean square was computed based on sequential shifts in pixel in x- and y-dimensions that were combined trigonometrically (*Figure 1—figure supplement 2, D*). Second, to detect large transient movement events, combined brain shift traces were filtered (Butterworth, at 1.5 Hz), and events larger than 5 μm (16 pixels) were detected as movement events (black dots in B).

## Behavioral analysis

### Licking detection

To track the movement of the tongue, videos of orofacial movement (65 Hz, side view) were processed using DeepLabCut (*Mathis et al., 2018*; *Figure 2—figure supplement 1*). The tip of the tongue, the location of the water spout and the position of the nose were labeled in randomly selected ~200 frames from six animals. In frames when the tongue was inside the mouth and was not visible, we estimated its location from the lips and jaw, instead of not labeling the tongue in these frames. This estimation prevented DeepLabCut from making a completely wrong guess in labeling the tongue for these frames.

The learning process was divided into three equal-duration periods. We confirmed that the division into six periods resulted in a saturating discrimination curve for anticipatory licking in the fifth and sixth periods (*Figure 3—figure supplement 1*). These last two periods in the six-period division correspond to the 'late phase' of the three-period division that we used.

### Running detection

The speed of treadmill was monitored as the output from a SpeedBelt apparatus (Phenosys). The locomotion period was defined as the duration in which the treadmill speed was above the median +0.5 x standard deviation for more than 200ms. Then, the initiation of the locomotion period was defined as a time point preceded by a non-locomotion period (when the running speed is below the threshold) of at least 0.5 s.

### Error analysis

To investigate how the cue discrimination of the animals impacts dopamine axonal activity, we separated the trials into those with correct discriminating behavior and those with incorrect behavior for reward and aversive conditions (*Figure 4A*). We used two types of discriminating behaviors, anticipatory licking and facial expressions (see below). We analyzed the late phase of the classical conditioning when animals showed robust anticipatory licking (*Figure 4—figure supplement 1*) or facial expressions (*Figure 4—figure supplement 2*). In one animal, anticipatory licking was not seen in the aversive condition (the fourth from the top of *Figure 4A*) during the late phase, so we analyzed the middle phase for that animal.

### Machine learning analysis of facial expressions, licking, and running

The anticipation of animals regarding upcoming unconditioned stimuli (reward or electrical shock) was quantified based on auditory predictive cues using a machine learning classifier (random forest classifier; *Breiman, 2001*).

Facial expressions were filmed by an infrared camera and analyzed with a random forest classifier combined with a deep neural network (*Figure 4—figure supplement 2*). First, features of facial expressions were extracted from a given temporal series of frames (i.e. a video) using a deep neural network model, the ResNet3D model. The ResNet3D model is a pretrained network consisting of 18 layers, optimized for videos and provided by PyTorch (*Tran et al., 2018*). The output from the final convolutional layer was fed into the random forest classifier. In our study, training was performed not on the pretrained ResNet3D, but on the random forest classifier. The random forest classifier was trained and tested with independent trials by fivefold cross-validation within each day. To prevent the random forest classifier from being overfit, only the top 400 features of the input were used, which were ranked by the F-value. To train the random forest classifier equally to the reward and aversive conditions despite their imbalanced frequency (seven or one out of eight trials, *Figure 3A*), an ensemble training technique was used (*Wallace et al., 2011*). the discrimination accuracy for reward and aversive conditions was computed separately and an average was taken with equal weights as a final discrimination accuracy. The equal weights prevented the accuracy computation from being dominated by the reward condition, which occurred more frequently than the aversive condition. To investigate the time course of the discrimination accuracy, accuracy computation was performed for a 500 ms time window instead of a 2 s window, and the window was systematically shifted by 160ms (*Figure 4—figure supplement 2*).

The discrimination accuracy based on anticipatory licking was also computed (*Figure 4—figure supplement 1*). To enable a comparison among facial features and licking, the random forest classifier was used. Instead of 400 features (facial expressions), the random forest classifier was fed with one feature (either the number of licking instances).

## Histology

Animals were perfused with 4% paraformaldehyde (PFA) in phosphate-buffered saline (PBS). GCaMP or tyrosine hydroxylase immunostaining was performed using standard procedures (*Figure 1C and D*). Coronal slices (thickness, 30 μm) were cut using a cryostat (Leica Microsystems) and blocked in carrier solution (5% bovine serum albumin; 0.3% Triton X-100 in 0.1 M PBS) for 2 hr at room temperature on a shaker. For GFP staining, slices were then incubated with anti-green fluorescent protein (GFP) primary antibody (anti-GFP, 1:1000, A11122, Invitrogen) for 18 hr at 4 °C on a shaker. After three rinses with 0.1 M PBS for 30 min, sections were incubated with Alexa-Fluor-488-conjugated donkey anti-rabbit secondary antibody (Invitrogen, 1:500 in carrier solution) for 1 hr at room temperature on a shaker. For tyrosine hydroxylase staining, additional incubation with anti- tyrosine hydroxylase (TH) primary antibody (anti-TH, 1:200, ab113) and Alexa-Fluor-568-conjugated donkey anti-sheep secondary antibody (Invitrogen, 1:500) was included. Cell nuclei were stained with DAPI (1:1000; D523, Dojindo). After a few additional rinses for 30 min in 0.1 M PBS were performed, slices were mounted on slide glasses for imaging. Images were acquired using a confocal laser-scanning microscopy (FV3000, Olympus) and a fluorescence microscope (VS200, Olympus).

## Experimental design and statistical analysis

Data are described as the median ± s.e.m. unless otherwise noted. Statistical significance was assessed using the non-parametric Wilcoxon signed-rank test, unless stated otherwise. Significance levels of data were denoted as * $p<0.05$, ** $p<0.01$ and *** $p<0.001$. $p>0.05$ was insignificant and was denoted as n.s.

## Lead contact

Further information and requests for reagents may be directed to the Lead Contact, Takashi R Sato ( satot@musc.edu).

## Materials availability

The plasmid construct has been deposited to Addgene (#216533).

## Acknowledgements

We thank J Zhang for technical help. This work was supported by grants from JSPS KAKENHI (20K16465) to YK, JSPS KAKENHI (20K16465) and JST PRESTO (JPMJPR2128) to KM, JSPS KAKENHI to TI (JP21K07459), BBRF Young Investigator Grant (29268), National Institute on Drug Abuse (COCA Pilot grant, P50DA046373), National Institute of Aging (R03 AG070517), National Institute of Neurological Disorders and Stroke (R21 NS125571, R01 NS131549), and NIH COBRE in Neurodevelopment and its Disorders (P20 GM148302) to TRS, and JST PRESTO (JPMJPR1883), JST FOREST (JPMJFR2245), NHMRC Ideas Grant (APP1184899), and KAKENHI (20K23378) to TKS.

## Additional information

### Funding

| Funder | Grant reference number | Author |
|---|---|---|
| Japan Society for the Promotion of Science | 20K16465 | Kei Majima |
| Japan Science and Technology Agency | 10.52926/jpmjpr2128 | Kei Majima |

| Funder | Grant reference number | Author |
|---|---|---|
| Japan Society for the Promotion of Science | JP21K07459 | Takahide Itokazu |
| BBRF Young Investigator Grant | 29268 | Takashi R Sato |
| National Institute on Drug Abuse | COCA Pilot grant P50 DA046373 | Takashi R Sato |
| National Institute of Aging | R03 AG070517 | Takashi R Sato |
| National Institute of Neurological Disorders and Stroke | R21 NS125571 | Takashi R Sato |
| National Institute of Neurological Disorders and Stroke | R01 NS131549 | Takashi R Sato |
| NIH COBRE in Neurodevelopment and its Disorders | P20 GM148302 | Takashi R Sato |
| Japan Science and Technology Agency | 10.52926/jpmjpr1883 | Tatsuo Sato |
| Japan Science and Technology Agency | JPMJFR2245 | Tatsuo Sato |
| National Health and Medical Research Council | APP1184899 | Tatsuo Sato |
| Japan Society for the Promotion of Science | 20K23378 | Tatsuo Sato |
| Japan Society for the Promotion of Science | 23K06151 | Yuki Kambe |

The funders had no role in study design, data collection and interpretation, or the decision to submit the work for publication.

## Author contributions

Kenta Abe, Zijing Hu, Data curation, Formal analysis; Yuki Kambe, Takahide Itokazu, Data curation; Kei Majima, Hideki Izumi, Takuma Tanaka, Software; Makoto Ohtake, Data curation, Methodology; Ali Momennezhad, Software, Formal analysis; Ashley Matunis, Emma Stacy, Formal analysis; Takashi R Sato, Tatsuo Sato, Conceptualization, Resources, Data curation, Software, Formal analysis, Supervision, Funding acquisition, Validation, Investigation, Visualization, Methodology, Writing – original draft, Project administration, Writing – review and editing

## Author ORCIDs

Kei Majima ⓘ http://orcid.org/0000-0002-2405-4113
Ali Momennezhad ⓘ http://orcid.org/0000-0002-8046-4387
Takashi R Sato ⓘ https://orcid.org/0000-0002-7623-9021
Tatsuo Sato ⓘ http://orcid.org/0000-0002-1279-5125

## Ethics

All experimental procedures were approved by local institutions supervising animal experiments at the Medical University of South Carolina (Permit Number: IACUC-2021-01236), Monash University (Permit Number: 18198), and Kagoshima University (Permit Number: MD22087).

Reviewer #1 (Public Review): https://doi.org/10.7554/eLife.91136.3.sa1
Reviewer #2 (Public Review): https://doi.org/10.7554/eLife.91136.3.sa2
Reviewer #3 (Public Review): https://doi.org/10.7554/eLife.91136.3.sa3
Author response https://doi.org/10.7554/eLife.91136.3.sa4

# Additional files

## Supplementary files
• MDAR checklist

## Data availability
The plasmid construct has been deposited to Addgene (#216533).All data and corresponding analysis codes reported in this study are available at GitHub. https://github.com/pharmedku/2024-elife-da-axon, copy archived at *pharmedku, 2024*.

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
