## [Editor Report · eLife assessment]

This **important** study shows that distinct midbrain dopaminergic axons in the medial prefrontal cortex respond to aversive and rewarding stimuli and suggest that they are biased toward aversive processing. The use of innovative microprism based two-photon calcium imaging to study single axon heterogeneity is **convincing**, although the experimental design makes it difficult to definitively distinguish aversive valence from stimulus salience in this dopamine projection. This work will be of interest to neuroscientists working on neuromodulatory systems, cortical function and decision making.

---

## [Referee Report · Reviewer #1 (Public Review)]

Summary:

In this manuscript, Abe and colleagues employ in vivo 2-photon calcium imaging of dopaminergic axons in the mPFC. The study reveals that these axons primarily respond to unconditioned aversive stimuli (US) and enhance their responses to initially-neutral stimuli after classical association learning. The manuscript is well-structured and presents results clearly. The utilization of a refined prism-based imaging technique, though not entirely novel, is well-implemented. The study's significance lies in its contribution to the existing literature by offering single-axon resolution functional insights, supplementing prior bulk measurements of calcium or dopamine release. Given the current focus on neuromodulator neuron heterogeneity, the work aligns well with current research trends and will greatly interest researchers in the field.

Comment on the revised version:

In my opinion, the authors did a great job with the revision of the manuscript.

---

## [Referee Report · Reviewer #2 (Public Review)]

Summary:

This study aims to address existing differences in the literature regarding the extent of reward versus aversive dopamine signaling in the prefrontal cortex. To do so, the authors chose to present mice with both a reward and an aversive stimulus during different trials each day. The authors used high spatial resolution two-photon calcium imaging of individual dopaminergic axons in the medial PFC to characterize the response of these axons to determine the selectivity of responses in unique axons. They also paired the reward (water) and an aversive stimulus (tail shock) with auditory tones and recorded across 12 days of associative learning.

The authors find that some axons respond to both reward and aversive unconditioned stimuli, but overall, there is a preference to respond to aversive stimuli consistent with expectations from prior studies that used other recording methods. The authors find that both of their two auditory stimuli initially drive responses in axons, but that with training axons develop more selective responses for the shock associated tone indicating that associative learning led to changes in these axon's responses. Finally, the authors use anticipatory behaviors during the conditioned stimuli and facial expressions to determine stimulus discrimination and relate dopamine axons signals with this behavioral evidence of discrimination. This study takes advantage of cutting-edge imaging approaches to resolve the extent to which dopamine axons in PFC respond appetitive or aversive stimuli. They conclude that there is a bias to respond to the aversive tail shock in most axons and weaker more sparse representation of water reward.

Strengths:

The strength of this study is the imaging approach that allows for investigation of the heterogeneity of response across individual dopamine axons unlike other common approaches such as fiber photometry which provide a measure of the average population activity. The use of appetitive and aversive stimuli to probe responses across individual axons is another strength as it reveals response diversity that is often overlooked in reward-only studies.

Weaknesses:

A weakness of this study is the design of the associative conditioning paradigm. The use of only a single reward and single aversive stimulus makes it difficult to know whether these results are specific to the valence of the stimuli versus the specific identity of the stimuli. Further, the reward presentations are more numerous than the aversive trials making it unclear how much novelty and habituation account for results. Moreover, the training seems somewhat limited by the low number of trials and did not result in strong associative conditioning. The lack of omission responses reported may reflect weak associative conditioning. Finally, the study provides a small advance in our understanding of dopamine signaling in the PFC and lacks evidence for if and what might be the consequence of these axonal responses on PFC dopamine concentrations and PFC neuron activity.

---

## [Referee Report · Reviewer #3 (Public Review)]

Summary:

The authors image dopamine axons in medial prefrontal cortex (mPFC) using microprism-mediated two-photon calcium imaging. They image these axons as mice learn that two auditory cues predict two distinct outcomes, tailshock, or water delivery. They find that some axons show a preference for encoding of the shock and some show a preference for encoding of water. The authors report a greater number of dopamine axons in mPFC that respond to shock. Across time, the shock-preferring axons begin to respond preferentially to the cue predicting shock, while there is a less pronounced increase in the water-responsive axons that acquire a response to the water-predictive cue (these axons also increase non-significantly to the shock-predictive cue). These data lead the authors to argue that dopamine axons in mPFC preferentially encode aversive stimuli.

Strengths:

The experiments are beautifully executed and the authors have mastered an impressively complex technique. Specifically, they are able to image and track individual dopamine axons in mPFC across days of learning. And this technique is used the way it should be: the authors isolate distinct dopamine axons in mPFC and characterize their encoding preferences and how this evolves across learning of cue-shock and cue-water contingencies. Thus, these experiments are revealing novel information about how aversive and rewarding stimuli is encoded at the level of individual axons, in a way that has not been done before. This is timely and important.

Weaknesses:

The overarching conclusion of the paper is that dopamine axons preferentially encode aversive stimuli. However, this is confounded by differences in the strength of the aversive and appetitive outcomes. As the authors point out, the axonal response to stimuli is sensitive to outcome magnitude (Supp Fig 3). That is, if you increase the magnitude of water or shock that is delivered, you increase the change in fluorescence that is seen in the axons. Unsurprisingly, the change in fluorescence that is seen to shock is considerably higher than water reward. Further, over 40% of the axons respond to water early in training [yet just a few lines below the authors write: "Previous studies have demonstrated that the overall dopamine release at the mPFC or the summed activity of mPFC dopamine axons exhibits a strong response to aversive stimuli (e.g., tail shock), but little to rewards", which seems inconsistent with their own data]. Given these aspects of the data, it could be the case that the dopamine axons in mPFC encodes different types of information and delegates preferential processing to the most salient outcome across time. The use of two similar sounding tones (9Khz and 12KHz) for the reward and aversive predicting cues are likely to enhance this as it requires a fine-grained distinction between the two cues in order to learn effectively. That is not to say that the mice cannot distinguish between these cues, rather that they may require additional processes to resolve the similarity, which are known to be dependent on the mPFC.

There is considerable literature on mPFC function across species that would support such a view. Specifically, theories of mPFC function (in particular prelimbic cortex, which is where the axon images are mostly taken) generally center around resolution of conflict in what to respond, learn about, and attend to. That is, mPFC is important for devoting the most resources (learning, behavior) to the most relevant outcomes in the environment. This data then, provides a mechanism for this to occur in mPFC. That is, dopamine axons signal to the mPFC the most salient aspects of the environment, which should be preferentially learnt about and responded towards. This is also consistent with the absence of a negative prediction error during omission: the dopamine axons show increases in responses during receipt of unexpected outcomes but do not encode negative errors. This supports a role for this projection in helping to allocate resources to the most salient outcomes and their predictors, and not learning per se. Below are a just few references from the rich literature on mPFC function (some consider rodent mPFC analogous to DLPFC, some mPFC), which advocate for a role in this region in allocating attention and cognitive resources to most relevant stimuli, and do not indicate preferential processing of aversive stimuli.

References:

1. Miller, E. K., & Cohen, J. D. (2001). An integrative theory of prefrontal cortex function. Annual review of neuroscience, 24(1), 167-202.

2. Bissonette, G. B., Powell, E. M., & Roesch, M. R. (2013). Neural structures underlying set-shifting: roles of medial prefrontal cortex and anterior cingulate cortex. Behavioural brain research, 250, 91-101.

3. Desimone, R., & Duncan, J. (1995). Neural mechanisms of selective visual attention. Annual review of neuroscience, 18(1), 193-222.

4. Sharpe, M. J., Stalnaker, T., Schuck, N. W., Killcross, S., Schoenbaum, G., & Niv, Y. (2019). An integrated model of action selection: distinct modes of cortical control of striatal decision making. Annual review of psychology, 70, 53-76.

5. Ridderinkhof, K. R., Ullsperger, M., Crone, E. A., & Nieuwenhuis, S. (2004). The role of the medial frontal cortex in cognitive control. science, 306(5695), 443-447.

6. Nee, D. E., Kastner, S., & Brown, J. W. (2011). Functional heterogeneity of conflict, error, task-switching, and unexpectedness effects within medial prefrontal cortex. Neuroimage, 54(1), 528-540.

7. Isoda, M., & Hikosaka, O. (2007). Switching from automatic to controlled action by monkey medial frontal cortex. Nature neuroscience, 10(2), 240-248.

---

## [Author Response]

The following is the authors’ response to the original reviews.

The reviewers praised multiple aspects of our study. Reviewer 1 noted that “the work aligns well with current research trends and will greatly interest researchers in the field.” Reviewer 2 highlighted the unique capability of our imaging approach, which “allows for investigation of the heterogeneity of response across individual dopamine axons, unlike other common approaches such as fiber photometry.” Reviewer 3 commented that “the experiments are beautifully executed” and “are revealing novel information about how aversive and rewarding stimuli is encoded at the level of individual axons, in a way that has not been done before.”

In addition to the positive feedback, the reviewers also provided useful criticisms and suggestions, some of which may not be fully addressed in a single study. For instance, questions regarding whether dopamine axons encode the valence or specific identity of the stimuli, or the most salient aspects of the environment, remain open. At the same time, as all the reviewers agreed, our report on the diversity of dopamine axonal responses using a novel imaging design introduces significant new insights to the neuroscience community. Following the reviewers’ recommendations, we have refrained from making interpretations that could be perceived as overinterpretation, such as concluding that “dopamine axons are involved in aversive processing.” This has necessitated extensive revisions, including modifying the title of our manuscript to make clear that the novelty of our work is revealing ‘functional diversity’ using our new imaging approach.

Below, we respond to the reviewers’ comments point by point.

**eLife assessment**
This valuable study shows that distinct midbrain dopaminergic axons in the medial prefrontal cortex respond to aversive and rewarding stimuli and suggest that they are biased toward aversive processing. The use of innovative microprism based two-photon calcium imaging to study single axon heterogeneity is solid, although the experimental design could be optimized to distinguish aversive valence from stimulus salience and identity in this dopamine projection. This work will be of interest to neuroscientists working on neuromodulatory systems, cortical function and decision making.
**Reviewer #1**
Summary:In this manuscript, Abe and colleagues employ in vivo 2-photon calcium imaging of dopaminergic axons in the mPFC. The study reveals that these axons primarily respond to unconditioned aversive stimuli (US) and enhance their responses to initially-neutral stimuli after classical association learning. The manuscript is well-structured and presents results clearly. The utilization of a refined prism-based imaging technique, though not entirely novel, is well-implemented. The study's significance lies in its contribution to the existing literature by offering single-axon resolution functional insights, supplementing prior bulk measurements of calcium or dopamine release. Given the current focus on neuromodulator neuron heterogeneity, the work aligns well with current research trends and will greatly interest researchers in the field.However, I would like to highlight that the authors could further enhance their manuscript by addressing study limitations more comprehensively and by providing essential details to ensure the reproducibility of their research. In light of this, I have a number of comments and suggestions that, if incorporated, would significantly contribute to the manuscript's value to the field.Strengths:Descriptive.Utilization of a well-optimized prism-based imaging method.Provides valuable single-axon resolution functional observations, filling a gap in existing literature.Timely contribution to the study of neuromodulator neuron heterogeneity.

We thank the reviewer for this positive assessment.

Weaknesses:(1) It's important to fully discuss the fact that the measurements were carried out only on superficial layers (30-100um), while major dopamine projections target deep layers of the mPFC as discussed in the cited literature (Vander Weele et al., 2018) and as illustrated in FigS1B,C. This limitation should be explicitly acknowledged and discussed in the manuscript, especially given the potential functional heterogeneity among dopamine neurons in different layers. This potential across-layer heterogeneity could also be the cause of discrepancy among past recording studies with different measurement modalities. Also, mentioning technical limitations would be informative. For example: how deep the authors can perform 2p-imaging through the prism? was the "30-100um" maximum depth the authors could get?

Thank you for pointing out this important issue about layer differences.

It is possible that the mesocortial pathway has layer-specific channels, with some neurons targeting supra granular layers and others targeting infragranular ones. Alternatively, it is also plausible that the axons of the same neurons branch into both superficial and deep layers. This is a critical issue that has not been investigated in anatomical studies and will require single-cell labeling of dopamine neurons (Matsuda et al 2009 and Aransay et al 2015). We now discuss this issue in the Discussion.

As for the imaging depth of 30–100 μm, we were unable to visualize deeper axons in a live view mode. Our imaging system has already been optimized to detect weak signals (e.g., we have employed an excitation wavelength of 980 nm, dispersion compensation, and a hybrid photodetector). It is possible that future studies using improved imaging approaches may be able to visualize deeper layers. Importantly, sparse axons in the supragranular layers are advantageous in detecting weak signals; dense labeling of axons would increase the background fluorescence relative to signals. We now reference this layer issue in the Results and Discussion sections.

(2) In the introduction, it seems that the authors intended to refer to Poulin et al. 2018 regarding molecular/anatomical heterogeneity of dopamine neurons, but they inadvertently cited Poulin et al. 2016 (a general review on scRNAseq). Additionally, the statement that "dopamine neurons that project to the PFC show unique genetic profiles (line 85)" requires clarification, as Poulin et al. 2018 did not specifically establish this point. Instead, they found at least the Vglut2/Cck+ population projects into mPFC, and they did not reject the possibility of other subclasses projecting to mPFC. Rather, they observed denser innervation with DAT-cre, suggesting that non-Vglut2/Cck populations would also project to mPFC. Discuss the potential molecular heterogeneity among mPFC dopamine axons in light of the sampling limitation mentioned earlier.

We thank the reviewer for pointing this out. Genetic profiles of PFC-projecting DA neurons are still being investigated, so describing them as “unique” was misleading. We have edited the Introduction accordingly, and now discuss this issue in detail in the Discussion.

(3) I find the data presented in Figure 2 to be odd. Firstly, the latency of shock responses in the representative axons (right panels of G, H) is consistently very long - nearly 500ms. It raises a query whether this is a biological phenomenon or if it stems from a potential technical artifact, possibly arising from an issue in synchronization between the 2-photon imaging and stimulus presentation. My reservations are compounded by the notable absence of comprehensive information concerning the synchronization of the experimental system in the method section.

The synchronization of the stimulus and data acquisition is accomplished at a sub-millisecond resolution. We use a custom-made MATLAB program that sends TTL commands to standard imaging software (ThorImage or ScanImage) and a stimulator for electrical shocks. All events are recorded as analogue inputs to a different DAQ to ensure synchronization. We have provided additional details regarding the configuration in the Methods section.

We consider that the long latency of shock response is biological. For instance, a similar long latency was found after electrical shock in a photometry imaging study (Kim, …, Deisseroth, 2016).

Secondly, there appear to be irregularities in Panel J. While the authors indicate that "Significant axons were classified as either reward-preferring (cyan) or aversive-preferring (magenta), based on whether the axons are above or below the unity line of the reward/aversive scatter plot (Line 566)," a cyan dot slightly but clearly deviates above the unity line (around coordinates (x, y) = (20, 21)). This needs clarification. Lastly, when categorizing axons for analysis of conditioning data in Fig3 (not Fig2), the authors stated "The color-coded classification (cyan/magenta) was based on k-means clustering, using the responses before classical conditioning (Figure 2J)". I do not understand why the authors used different classification methods for two almost identical datasets.

We thank the reviewer for pointing out these insufficient descriptions. We classified the axons using k-means clustering, and the separation of the two clusters happened to roughly coincide with the unity line of the reward/aversive scatter plot in Fig 2J. In other words, we did not use the unity line to classify the data points (which is why the color separation of the histogram is not at 45 degrees). We have clarified this point in the Methods section.

(4) In connection with Point 3, conducting separate statistical analyses for aversive and rewarding stimuli would offer a fairer approach. This could potentially reveal a subset of axons that display responses to both aversive and appetitive stimuli, aligning more accurately with the true underlying dynamics. Moreover, the characterization of Figure 2J as a bimodal distribution while disregarding the presence of axons responsive to both aversive and appetitive cues seems somewhat arbitrary and circular logic. A more inclusive consideration of this dual-responsive population could contribute to a more comprehensive interpretation.

We also attempted k-means clustering with additional dimensions (e.g., temporal domains as shown in Fig. 3I, J), but no additional clusters were evident. We note that the lack of other clusters does not exclude the possibility of their existence, which may only become apparent with a substantial increase in the number of samples. In the current report, we present the clusters that were the easiest/simplest for us to identify.

Additionally, we have revised our manuscript to reflect that many axons respond to both reward and aversive stimuli, and that aversive-preferring axons do not exclusively respond to the aversive stimulus.

(5) The contrast in initialization to novel cues between aversive and appetitive axons mirrors findings in other areas, such as the tail-of-striatum (TS) and ventral striatum (VS) projecting dopamine neurons (Menegas et al., 2017, not 2018). You might consider citing this very relevant study and discussing potential collateral projections between mPFC and TS or VS.

Thank you for pointing this out. We have now included Menegas et al., 2017, and also discuss the possibility of collaterals to these areas. In addition, we also referred to Azcorra et al., 2023 - this was published after our initial submission.

(6) The use of correlation values (here >0.65) to group ROIs into axons is common but should be justified based on axon density in the FOV and imaging quality. It's important to present the distribution of correlation values and demonstrate the consistency of results with varying cut-off values. Also, provide insights into the reliability of aversive/appetitive classifications for individual ROIs with high correlations. Importantly, if you do the statistical testing and aversive/appetitive classifications for individual ROIs with above-threshold high correlation (to be grouped into the same axon), do they always fall into the same category? How many false positives/false negatives are observed?"Our results remained similar for different correlation threshold values (Line 556)" (data not shown) is obsolete.

We have conducted additional analysis using correlation values 0.5 and 0.3 that resulted in a smaller number of axon terminals. In essence, the relationship between reward responses and aversive responses remained very similar to Fig. 2J, K.

**Author response image 1. sa4fig1:** 

**Reviewer #2 (Public Review):**
Summary:This study aims to address existing differences in the literature regarding the extent of reward versus aversive dopamine signaling in the prefrontal cortex. To do so, the authors chose to present mice with both a reward and an aversive stimulus during different trials each day. The authors used high spatial resolution two-photon calcium imaging of individual dopaminergic axons in the medial PFC to characterize the response of these axons to determine the selectivity of responses in unique axons. They also paired the reward (water) and an aversive stimulus (tail shock) with auditory tones and recorded across 12 days of associative learning.The authors find that some axons respond to both reward and aversive unconditioned stimuli, but overall, there is a strong preference to respond to aversive stimuli consistent with expectations from prior studies that used other recording methods. The authors find that both of their two auditory stimuli initially drive responses in axons, but that with training axons develop more selective responses for the shock associated tone indicating that associative learning led to changes in these axon's responses. Finally, the authors use anticipatory behaviors during the conditioned stimuli and facial expressions to determine stimulus discrimination and relate dopamine axons signals with this behavioral evidence of discrimination. This study takes advantage of cutting-edge imaging approaches to resolve the extent to which dopamine axons in PFC respond appetitive or aversive stimuli. They conclude that there is a strong bias to respond to the aversive tail shock in most axons and weaker more sparse representation of water reward.Strengths:The strength of this study is the imaging approach that allows for investigation of the heterogeneity of response across individual dopamine axons, unlike other common approaches such as fiber photometry which provide a measure of the average population activity. The use of appetitive and aversive stimuli to probe responses across individual axons is another strength.

We thank the reviewer for this positive assessment.

Weaknesses:A weakness of this study is the design of the associative conditioning paradigm. The use of only a single reward and single aversive stimulus makes it difficult to know whether these results are specific to the valence of the stimuli versus the specific identity of the stimuli. Further, the reward presentations are more numerous than the aversive trials making it unclear how much novelty and habituation account for results. Moreover, the training seems somewhat limited by the low number of trials and did not result in strong associative conditioning. The lack of omission responses reported may reflect weak associative conditioning. Finally, the study provides a small advance in our understanding of dopamine signaling in the PFC and lacks evidence for if and what might be the consequence of these axonal responses on PFC dopamine concentrations and PFC neuron activity.

We thank the reviewer for the suggestions.

We agree that interpreting the response change during classical conditioning is not straightforward. Although the reward and aversive stimuli we employed are commonly used in the field, future studies with more sophisticated paradigms will be necessary to address whether dopamine axons encode the valence of the stimuli, the specific identity of the stimuli, or novelty and habituation. In our current manuscript, we refrain from making a conclusion that distinct groups of neurons encode different valances. In fact, many axons respond to both stimuli, at different ratios. We have removed descriptions that may suggest exclusive coding of reward or aversive processing. Additionally, we have extensively discussed possible interpretations.

In terms of the strength of the conditioning association, behavioral results indicated that the learning plateaued – anticipatory behaviors did not increase during the last two phases when the conditioned span was divided into six phases (Figure 3–figure supplement 1).

Our goal in the current manuscript is to provide new insight into the functional diversity of dopamine axons in the mPFC. Investigating the impact of dopamine axons on local dopamine concentration and neural activity in the mPFC is important but falls beyond the scope of our current study. In particular, given the functional diversity of dopamine axons, interpreting bulk optogenetic or chemogenetic axonal manipulation experiments would not be straightforward. As suggested, measuring the dopamine concentration through two-photon imaging of dopamine sensors and monitoring the activity of dopamine recipient neurons (e.g., D1R- or D2R-expressing neurons) is a promising approach that we plan to undertake in the near future.

**Reviewer #3 (Public Review):**
Summary:The authors image dopamine axons in medial prefrontal cortex (mPFC) using microprism-mediated two-photon calcium imaging. They image these axons as mice learn that two auditory cues predict two distinct outcomes, tailshock or water delivery. They find that some axons show a preference for encoding of the shock and some show a preference for encoding of water. The authors report a greater number of dopamine axons in mPFC that respond to shock. Across time, the shock-preferring axons begin to respond preferentially to the cue predicting shock, while there is a less pronounced increase in the water-responsive axons that acquire a response to the water-predictive cue (these axons also increase non-significantly to the shock-predictive cue). These data lead the authors to argue that dopamine axons in mPFC preferentially encode aversive stimuli.Strengths:The experiments are beautifully executed and the authors have mastered an impressively complex technique. Specifically, they are able to image and track individual dopamine axons in mPFC across days of learning. This technique is used the way it should be: the authors isolate distinct dopamine axons in mPFC and characterize their encoding preferences and how this evolves across learning of cue-shock and cue-water contingencies. Thus, these experiments are revealing novel information about how aversive and rewarding stimuli is encoded at the level of individual axons, in a way that has not been done before. This is timely and important.

We thank the reviewer for this positive assessment.

Weaknesses:The overarching conclusion of the paper is that dopamine axons preferentially encode aversive stimuli. This is prevalent in the title, abstract, and throughout the manuscript. This is fundamentally confounded. As the authors point out themselves, the axonal response to stimuli is sensitive to outcome magnitude (Supp Fig 3). That is, if you increase the magnitude of water or shock that is delivered, you increase the change in fluorescence that is seen in the axons. Unsurprisingly, the change in fluorescence that is seen to shock is considerably higher than water reward.

We agree that the interpretation of our results is not straightforward. Our current manuscript now focuses on our strength, which is reporting the functional diversity of dopamine axons. Therefore, we avoid using the word ‘encode’ when describing the response.

We believe that our results could reconcile the apparent discrepancy as to why some previous studies reported only aversive responses while others reported reward responses. In particular, if the reward volume were very small, the reward response could go undetected.

Further, when the mice are first given unexpected water delivery and have not yet experienced the aversive stimuli, over 40% of the axons respond [yet just a few lines below the authors write: "Previous studies have demonstrated that the overall dopamine release at the mPFC or the summed activity of mPFC dopamine axons exhibits a strong response to aversive stimuli (e.g., tail shock), but little to rewards", which seems inconsistent with their own data].

We always recorded the reward and aversive response together, which might have confused the reviewer. Therefore, there is no inconsistency in our data. We have clarified our methods and reasoning accordingly.

Given these aspects of the data, it could be the case that the dopamine axons in mPFC encodes different types of information and delegates preferential processing to the most salient outcome across time.

This is certainly an exciting interpretation, so we have included it in our discussion. Meanwhile, ‘the most salient outcome’ alone cannot fully capture the diverse response patterns of the dopaminergic axons, particularly reward-preferring axons. We discuss our findings in more detail in the revised manuscript.

The use of two similar sounding tones (9Khz and 12KHz) for the reward and aversive predicting cues are likely to enhance this as it requires a fine-grained distinction between the two cues in order to learn effectively. There is considerable literature on mPFC function across species that would support such a view. Specifically, theories of mPFC function (in particular prelimbic cortex, which is where the axon images are mostly taken) generally center around resolution of conflict in what to respond, learn about, and attend to. That is, mPFC is important for devoting the most resources (learning, behavior) to the most relevant outcomes in the environment. This data then, provides a mechanism for this to occur in mPFC. That is, dopamine axons signal to the mPFC the most salient aspects of the environment, which should be preferentially learned about and responded towards. This is also consistent with the absence of a negative prediction error during omission: the dopamine axons show increases in responses during receipt of unexpected outcomes, but do not encode negative errors. This supports a role for this projection in helping to allocate resources to the most salient outcomes and their predictors, and not learning per se. Below are a just few references from the rich literature on mPFC function (some consider rodent mPFC analogous to DLPFC, some mPFC), which advocate for a role in this region in allocating attention and cognitive resources to most relevant stimuli, and do not indicate preferential processing of aversive stimuli.

Distinguishing between 9 kHz and 12 kHz sound tones may not be that difficult, considering anticipatory licking and running are differentially manifested. In addition, previous studies have shown that mice can distinguish between two sound tones when they are separated by 7% (de Hoz and Nelken 2014). Nonetheless, we agree with the attractive interpretation that “the mPFC devotes the most resources (learning, behavior) to the most relevant outcomes in the environment” and that dopamine is a mechanism for this. Therefore, we discuss this interpretation in the revised text.

References:

(1) Miller, E. K., & Cohen, J. D. (2001). An integrative theory of prefrontal cortex function. Annual review of neuroscience, 24(1), 167-202.

(2) Bissonette, G. B., Powell, E. M., & Roesch, M. R. (2013). Neural structures underlying set-shifting: roles of medial prefrontal cortex and anterior cingulate cortex. Behavioural brain research, 250, 91101.

(3) Desimone, R., & Duncan, J. (1995). Neural mechanisms of selective visual attention. Annual review of neuroscience, 18(1), 193-222.

(4) Sharpe, M. J., Stalnaker, T., Schuck, N. W., Killcross, S., Schoenbaum, G., & Niv, Y. (2019). An integrated model of action selection: distinct modes of cortical control of striatal decision making. Annual review of psychology, 70, 53-76.

(5) Ridderinkhof, K. R., Ullsperger, M., Crone, E. A., & Nieuwenhuis, S. (2004). The role of the medial frontal cortex in cognitive control. science, 306(5695), 443-447.

(6) Nee, D. E., Kastner, S., & Brown, J. W. (2011). Functional heterogeneity of conflict, error, taskswitching, and unexpectedness effects within medial prefrontal cortex. Neuroimage, 54(1), 528-540.

(7) Isoda, M., & Hikosaka, O. (2007). Switching from automatic to controlled action by monkey medial frontal cortex. Nature neuroscience, 10(2), 240-248.

**Reviewer #1 (Recommendations For The Authors):**
Specific Suggestions and Questions on the Methods Section:In general, the methods part is not well documented and sometimes confusing. Thus, as it stands, it hinders reproducible research. Specific suggestions/questions are listed in the following section.(1) Broussard et al. 2018 introduced axon-GCaMP6 instead of axon-jGCaMP8m. The authors should provide details about the source of this material. If it was custom-made, a description of the subcloning process would be appreciated. Additionally, consider depositing sequence information or preferably the plasmid itself. Furthermore, the introduction of the jGCaMP8 series by Zhang, Rozsa, et al. 2023 should be acknowledged and referenced in your manuscript.

We thank the reviewer for pointing this out. We have now included details on how we prepared the axon-jGCaMP8m, which was based on plasmids available at Addgene. Additionally, we have deposited our construct to Addgene ( https://www.addgene.org/216533/ ). We have also cited Janelia’s report on jGCaMP8, Zhang et al.

(2) The authors elaborate on the approach taken for experimental synchronization. Specifically, how was the alignment achieved between 2-photon imaging, treadmill recordings, aversive/appetitive stimuli, and videography? It would be important to document the details of the software and hardware components employed for generating TTLs that trigger the pump, stimulator, cameras, etc.

We have now included a more detailed explanation about the timing control. We utilize a custommade MATLAB program that sends TTL square waves and analogue waves via a single National Instruments board (USB-6229) to control two-photon image acquisition, behavior camera image acquisition, water syringe movement, current flow from a stimulator, and sound presentation. We also continuously recorded at 30 kHz via a separate National Instrument board (PCIe-6363) the frame timing of two-photon imaging, the frame timing of a behavior camera, copies of command waves (sent to the syringe pump, the stimulator, and the speaker), and signals from the treadmill corresponding to running speed.

(3) The information regarding the cameras utilized in the study presents some confusion. In one instance, you mention, "To monitor licking behavior, the face of each mouse was filmed with a camera at 60 Hz (CM3-U3-13Y3M-CS, FLIR)" (Line 488). However, there's also a reference to filming facial expressions using an infrared web camera (Line 613). Could you clarify whether the FLIR camera (which is an industrial CMOS not a webcam) is referred to as a webcam? Alternatively, if it's a different camera being discussed, please provide product details, including pixel numbers and frame rate for clarity.

We thank the reviewer for pointing this out. This was a mistake on our end. The camera used in the current project was a CM3-U3-13Y3M-CS, not a web camera. We have now corrected this.

(4) Please provide more information about the methodology employed for lick detection. Specifically, did the authors solely rely on videography for this purpose? If so, why was an electrical (or capacitive) detector not used? It would provide greater accuracy in detecting licking.

Lick detection was performed offline based on videography, using DeepLabCut. As licking occurs at a frequency of ~6.5 Hz (Xu, …, O’Connor Nature Neurosci, 2022), the movement can be detected at a frame rate of 60 Hz. Initially, we used both a lick sensor and videography. However, we favored videography because it could potentially provide non-binary information.

Other Minor Points:(5) Ensure consistency in the citation format; both Vander Weele et al. 2018 and Weele et al. 2019, share the same first author.

Thank you for pointing this out. Endnote processes the first author’s name differently depending on the journal. We fixed the error manually. The first paper (2018) is an original research paper, and the second one (2019) is a review about how dopamine modulates aversive processing in the mPFC. We cited the second one in three instances where we mentioned review papers.

(6) The distinction between "dashed vs dotted lines" in Figure 3K and 3M appears to be very confusing. Please consider providing a clearer visualization/labeling to mitigate this confusion.

We have now changed the line styles.

(7) Additionally plotting mean polar angles of aversive/appetitive axons as vectors in the Cartesian scatter plots (2J, 3I,J) would make interpretation easier.

We have now made this change to Figures 2, 3, 4.

(8) Data and codes should be shared in a public database. This is important for reproducible research and we believe that "available from the corresponding author upon reasonable request" is outdated language.

We have uploaded the data to GitHub, https://github.com/pharmedku/2024-elife-da-axon.

**Reviewer #2 (Recommendations For The Authors):**
(1) Authors don't show which mouse each axon data comes from making it hard to know if differences arise from inter-mouse differences vs differences in axons. The best way to address this point is to show similar plots as Figure 2J & K but broken down by mouse to shows whether each mouse had evidence of these two clusters.

We have now made this change to Figure 2-figure supplement 3.

(2) Line 166: Should this sentence point to panels 2F, G, H rather than 2I which doesn't show a shock response?

We thank the reviewer for pointing this out. We have fixed the incorrect labels.

Line 195: The population level bias to aversive stimuli was shown previously using photometry so it is not justified to say "for the first time" regarding this statement.

We have adjusted this sentences so the claim of ”for the first time” is not associated with the population-level bias.

(4) The paper lacks a discussion of the potential role that novelty plays in the amplitude of the responses given that tail shocks occur less often that rewards. Is the amplitude of the first reward of the day larger than subsequent rewards? Would tail shock responses decay if they occurred in sequential trials?

Following the reviewer's suggestion, we conducted a comparison of individual axonal responses to both conditioned and unconditioned stimuli across the first trial and subsequent trials. Our findings reveal a notable trend: aversive-preferring axons exhibited attenuation in response to CSreward, yet enhancement in response to CSaversive. Conversely, the response of these axons to USreward was attenuated, with no significant change observed for USaversive. In contrast, reward-preferring axons displayed an invariable activity pattern from the initial trial, highlighting the functional diversity present within dopamine axons. This analysis has been integrated into Figure 3-figure supplement 4 and is elaborated upon in the Discussion section.

(5) Fix typo in Figure 1 - supplement 1. *Shift*

We have now corrected this. Thank you.

(6) The methods section needs information about trial numbers. Please indicate how many trials were presented to each mouse per day.

We have now added the information about trial numbers to the Methods section.

**Reviewer #3 (Recommendations For The Authors):**
In line with the public review, my recommendation is for the authors to remain as objective about their data as possible. There are many points in the manuscript where the authors seem to directly contradict their own data. For example, they first detail that dopamine axons respond to unexpected water rewards. Indeed, they find that there are 40% of dopamine axons that respond in this way. Then, a few paragraphs later they state: "Previous studies have demonstrated that the overall dopamine release at the mPFC or the summed activity of mPFC dopamine axons exhibits a strong response to aversive stimuli (e.g., tail shock), but little to rewards". As detailed above, I do not think these data support an idea that dopamine axons in mPFC preferentially encode aversive outcomes. If the authors wanted to examine a role for mPFC in preferential encoding of aversive stimuli, you would first have to equate the outcomes by magnitude and then compare how the axons acquire preferences across time. Alternatively, a prediction of a more general process that I detail above would predict that you could give mice two rewards that differ in magnitude (e.g., lots of food vs. small water) and you would see the same results that the authors have seen here (i.e., a preference for the food, which is the larger and more salient outcome). Without other tests of how dopamine axons in mPFC respond to situations like this, I don't think any conclusion around mPFC in favoring aversive stimuli can be made.

As suggested, we have made the current manuscript as objective as possible, removing interpretation aspects regarding what dopamine axons encode and emphasizing their functional diversity. In particular, we remove the word ‘encode’ when describing the response of dopamine axons.

Although it may have appeared unclear, there was no contradiction within our data regarding the response to reward and aversive stimuli. We have now improved the readability of the Results and Methods sections. Concerning the interpretation of what exactly the mPFC dopamine axons encode, we have rewritten the discussion to be as objective about our data as possible, as suggested. We also have edited our title and abstract accordingly. Meanwhile, we wish to emphasize that our reward and aversive stimuli are standard paradigms commonly used in the field. We believe, and all the reviewers agreed, that reporting the diversity of dopamine axonal responses with a novel imaging design constitutes new insight for the neuroscience community. Therefore, we have decided to leave the introduction of new behavioral tasks for future studies and instead expanded our discussion.

As mentioned, I think the experiments are executed really well and the technological aspects of the authors' methods are impressive. However, there are also some aspects of the data presentation that would be improved. Some of the graphs took a considerable amount of effort to unpack. For example, Figure 4 is hard going. Is there a way to better illustrate the main points that this figure wants to convey? Some of this might be helped by a more complete description in the figure captions about what the data are showing. It would also be great to see how the response of dopamine axons changes across trial within a session to the shock and water-predictive cues. Supp Figure 1 should be in the main text with standard error and analyses across time. Clarifying these aspects of the data would make the paper more relevant and accessible to the field.

We thank the reviewer for pointing out that the legend of Figure 4 was incomplete. We have fixed it, along with improving the presentation of the figure. We have also prepared a new figure (Figure 3– figure supplement 4) to compare CSaversive and CSreward signals for the first and rest of the trials within daily sessions, revealing further functional diversity in dopamine axons. We have decided to keep Figure 1–figure supplement 2 as a figure supplement with an additional analysis, as another reviewer pointed out that the design is not completely new. Furthermore, as eLife readers can easily access figure supplements, we believe it is appropriate to maintain it in this way.

Minor points:(1) What is the control period for the omission test? Was omission conducted for the shock?

The control period for reward omission is a 2-second period just before the CS onset. We did not include shock omission, because a sufficient number of trials (> 6 trials) for the rare omission condition could not be achieved within a single day.

(2) The authors should mention how similar the tones were that predicted water and shock.

According to de Hoz and Nelken (2014), a frequency difference of 4–7% is enough for mice to discriminate between tones. In addition, anticipatory licking and running confirmed that the mice could discriminate between the frequencies. We have now included this information in the Discussion.

(3) I realize the viral approach used in the current studies may not allow for an idea of where in VTA dopamine neurons are that project to mPFC- is there data in the literature that speak to this? Particularly important as we now know that there is considerable heterogeneity in dopamine neuronal responses, which is often captured by differences in medial/lateral position within VTA.

Some studies have suggested that mesocortical dopamine neurons are located in the medial posterior VTA (e.g., Lammel et al., 2008). However, in mouse anterograde tracing, it is not possible to spatially confine the injection of conventional viruses/tracers. We now refer to Lammel et al., 2008 in the Introduction.